# Homology sensing via non-linear amplification of sequence-dependent pausing by RecQ helicase

Yeonee Seol[1†], Gábor M Harami[2†], Mihály Kovács[2,3]*, Keir C Neuman[1]*

[1]Laboratory of Single Molecule Biophysics, National Heart, Lung, and Blood Institute, National Institutes of Health, Bethesda, United States; [2]Department of Biochemistry, ELTE-MTA "Momentum" Motor Enzymology Research Group, Eötvös Loránd University, Budapest, Hungary; [3]Department of Biochemistry, MTA-ELTE Motor Pharmacology Research Group, Eötvös Loránd University, Budapest, Hungary

**Abstract** RecQ helicases promote genomic stability through their unique ability to suppress illegitimate recombination and resolve recombination intermediates. These DNA structure-specific activities of RecQ helicases are mediated by the helicase-and-RNAseD like C-terminal (HRDC) domain, via unknown mechanisms. Here, employing single-molecule magnetic tweezers and rapid kinetic approaches we establish that the HRDC domain stabilizes intrinsic, sequence-dependent, pauses of the core helicase (lacking the HRDC) in a DNA geometry-dependent manner. We elucidate the core unwinding mechanism in which the unwinding rate depends on the stability of the duplex DNA leading to transient sequence-dependent pauses. We further demonstrate a non-linear amplification of these transient pauses by the controlled binding of the HRDC domain. The resulting DNA sequence- and geometry-dependent pausing may underlie a homology sensing mechanism that allows rapid disruption of unstable (illegitimate) and stabilization of stable (legitimate) DNA strand invasions, which suggests an intrinsic mechanism of recombination quality control by RecQ helicases.

DOI: https://doi.org/10.7554/eLife.45909.001

*For correspondence:
mihaly.kovacs@ttk.elte.hu (MK);
neumankc@mail.nih.gov (KCN)

†These authors contributed equally to this work

Competing interests: The authors declare that no competing interests exist.

## Introduction

RecQ helicases are a family of DNA helicases that play essential roles in maintaining genomic integrity through extensive involvement in DNA recombination, replication, and repair pathways (*Bachrati and Hickson, 2003*; *Bennett and Keck, 2004*; *Chu and Hickson, 2009*). *Escherichia coli* RecQ (*Ec* RecQ) helicase is the founding member of the family (*Nakayama et al., 1984*) and plays roles in both suppressing illegitimate recombination and facilitating various steps of DNA recombinational repair (*Hanada et al., 1997*; *Ryder et al., 1994*; *León-Ortiz et al., 2018*). RecQ helicases are highly conserved from bacteria to humans and eukaryotic RecQ helicases have been shown to play similar pro- and anti-recombination functions. Most unicellular organisms, such as *E. coli* and yeast, express a single RecQ homolog, whereas multi-cellular organisms often possess multiple RecQ helicases specialized to different roles in genome maintenance processes.

The fundamental conserved activity of RecQ helicases is the ATP-dependent unwinding of double-stranded DNA (*Nakayama et al., 1984*). All RecQ members possess two evolutionarily conserved RecA-like helicase domains with an ATP binding and hydrolysis site located in a cleft between them (*Bennett and Keck, 2004*; *Chu and Hickson, 2009*; *Bachrati and Hickson, 2008*). Similar to other superfamily (SF) one and SF2 helicases, RecQ members also contain N- and C-terminal accessory domains that provide additional or specialized functionalities (*Fairman-Williams et al.,*

**eLife digest** Molecules of DNA carry instructions for all of the biological processes that happen in cells. Therefore, it is very important that cells maintain their DNA and quickly repair any damage. DNA molecules are made of two strands that twist together to form a double helix. The most reliable way to repair damage affecting both DNA strands involves a process known as homologous recombination. In this process, one of the strands of the broken DNA joins up with a strand of an identical or similar DNA molecule to make a triple-stranded structure known as a D-loop. This allows the cell to rebuild the damaged DNA using the intact DNA as a template.

To ensure that the DNA is repaired correctly, enzymes known as RecQ helicases bind to and unwind D-loops if the strand pairs are poorly matched, whilst not disrupting pairs that are correctly matched. It remains unclear, however, how these enzymes are able to distinguish whether DNA strands in D-loops are a good or bad pair.

To address this question, Seol, Harami et al. measured how individual RecQ helicases from a bacterium known as *Escherichia coli* unwind DNA. The experiments showed that the enzymes were better able to unwind sections of double-stranded DNA that were less stable than other sections of DNA (indicating the two strands may be a bad match). This causes the helicase to pause at stable sections of the DNA as it unwinds the double helix of the D-loop. Further experiments showed that a region of the helicase known as the HRDC domain increased the duration of these pauses, leading to a dramatic decrease in the unwinding speed.

Seol, Harami et al. propose that this difference in unwinding speed prevents RecQ from unwinding legitimate matching D-loops while permitting rapid disruption of illegitimate D-loops that could lead to damaged DNA being repaired incorrectly. Mutations in the human versions of RecQ helicases lead to Bloom's syndrome and Werner's syndrome in which individuals are predisposed to developing cancer. Understanding how cells repair DNA may ultimately help to treat individuals with these and other similar conditions.

DOI: https://doi.org/10.7554/eLife.45909.002

*2010*). The RecQ C-terminal domain (RQC) comprises zinc binding and winged-helix (WH) sub-domains associated with protein structural integrity and duplex DNA binding, respectively. Although less conserved, many RecQ-family members, including *Ec* RecQ and multiple human RecQ homo-logs, possess an accessory single-stranded (ss) DNA-binding module termed the helicase-and-RNAseD-C-terminal (HRDC) domain (*Bernstein and Keck, 2005*; *Vindigni and Hickson, 2009*). The HRDC, while generally dispensable for helicase activity, is critical for certain recombination interme-diate processing steps, such as disruption of displacement strand (D-loop) invasion and double Holli-day junction resolution (*Rezazadeh, 2012*; *Singh et al., 2012*; *Chatterjee et al., 2014*; *Harami et al., 2017*). Biochemical studies have established that full length RecQ has a higher ssDNA binding affinity than RecQ constructs lacking the HRDC, which is consistent with the findings that the interaction between the HRDC and ssDNA contributes to DNA substrate specificity of RecQ heli-cases (*Bernstein and Keck, 2005*; *Vindigni and Hickson, 2009*).

Recently, we provided evidence that HRDC interactions contribute to DNA substrate-geometry dependent binding orientation and unwinding by RecQ, and demonstrated that these HRDC-medi-ated interactions play a role in suppressing illegitimate recombination in *E. coli* (*Harami et al., 2017*). Whereas these findings indicate that the HRDC strongly favors binding of RecQ to D-loop structures in an orientation that promotes disruption of the invading DNA strand (*Harami et al., 2017*), it is not clear how RecQ can subsequently discriminate between homologous and non-homol-ogous strand invasions; once correctly oriented on the D-loop, RecQ can unwind any invading strand and indiscriminately disrupt all D-loop formations. In this work we identify a potential solution to this quandary, suggested by the observation that HRDC-dependent pausing during hairpin DNA unwind-ing is not random but occurs repeatedly at distinct positions on the DNA hairpin. We reason that if the frequency or duration of the unwinding pauses is related to the degree of DNA homology, then the more than 10-fold decrease in average unwinding rate due to pausing can provide a mechanism of homology sensing. Thus, if pausing is correlated with homology, then the resulting modulation of

the average unwinding rate of an oriented RecQ helicase will result in discrimination of legitimate versus illegitimate D-loops.

To test this theory, we set out to determine the origin of HRDC-mediated pausing by investigating the unwinding mechanism of *E. coli* RecQ and HRDC-induced pausing using single-molecule magnetic tweezers (MT)-based assays and rapid transient kinetic assays. We found that long-lived HRDC-induced pauses of wild type RecQ (RecQ$^{wt}$) and shorter-lived pauses of RecQ core domain (HRDC deletion mutant; RecQ-dH) are sequence dependent and both correlate with DNA duplex stability. Sequence-dependent pausing is a direct consequence of the unique DNA unwinding mechanism: RecQ unwinds one base-pair per ATP hydrolysis cycle but releases the nascent ssDNA only after unwinding ~5 bp. The translocation kinetics arising from this 5 bp kinetic step depend on the duplex stability, which results in sequence-dependent pausing of the core RecQ that is further stabilized by the HRDC binding to the displaced ssDNA. Kinetic modeling indicates that the affinity of the HRDC for ssDNA is enhanced at pause sites, rather than remaining constant. The HRDC thus acts as a non-linear amplifier of the transient sequence-dependent pauses of the core enzyme. Our study demonstrates that the coupling between the core unwinding mechanism and the HRDC-ssDNA interactions dramatically alter the mode of unwinding in a sequence dependent manner, and, in conjunction with previous work, potentially implicates a mechanistic basis for recombination quality control provided by RecQ helicases.

## Results

### RecQ pause positions are strongly correlated with DNA sequence

Single-molecule measurements of RecQ helicase unwinding activity were performed with 174- or 584-base pair DNA hairpins using an MT apparatus (*Figure 1A*). DNA hairpin substrates were attached to the flow-cell surface and to a 1- or 2.8 µm magnetic bead via a 1.1 kbp double-stranded DNA handle and 60-nucleotides of single-stranded poly-dT, respectively (*Figure 1A*). Measurements with the DNA hairpin were performed at a constant force of 8 pN under which the hairpin did not open spontaneously. In the presence of RecQ helicase (20–100 pM), unwinding activity was monitored in real-time by tracking the three-dimensional position of a tethered bead at 60 or 200 Hz. Trajectories of the bead extension as a function of time were analyzed by fitting with a *T*-test based step finding algorithm to obtain the mean unwinding rate, the 'step' unwinding rate between pauses, the pause positions, and the pause durations (*Harami et al., 2017*; *Seol et al., 2016*).

As described previously (*Harami et al., 2017*), frequent pausing and strand-switching by WT RecQ (RecQ$^{WT}$) is caused by the HRDC as the HRDC deletion mutant (RecQ-dH) shows significantly less pausing during DNA hairpin unwinding (*Figure 1B*). Pausing is attributed to transient binding of the HRDC domain to the displaced single-stranded DNA behind RecQ. Since both the displaced and the translocation strands of ssDNA are under tension in the hairpin substrate, binding of the HRDC to the displaced strand will prevent forward motion of the helicase. HRDC binding to either duplex DNA ahead of the helicase, or to the translocation strand of ssDNA behind the helicase, are ruled out by the lack of pauses during the unwinding of a 'gapped' DNA substrate in which the displaced strand is not constrained (*Harami et al., 2017*). Given the mechanical origin of the pausing associated with transient binding of the HRDC, the pause positions would be expected to be random, dependent on the stochastic kinetics of the interaction between HRDC and the displaced ssDNA.

Interestingly, the dwell-time histogram as a function of position for RecQ$^{WT}$ unwinding traces exhibits peaks at distinct positions along the hairpin (*Figure 2A*; top). The peaks in the dwell-time histogram of unwinding traces arise from long and/or frequent pauses at specific positions during DNA hairpin unwinding by RecQ helicase (*Figure 2—figure supplement 1*). To identify the sequence context of the pauses, the extension change associated with DNA hairpin opening was converted to base-pairs via the worm-like chain (WLC) model of DNA (*Manosas et al., 2010*). Each unwound base pair resulted in the increase of the molecular extension by two ssDNA nucleotides, which at an applied force of 8 pN corresponds to ~0.8 nm assuming a 1 nm persistence length and a 0.65 nm inter-phosphate distance. With this conversion factor, the extension change for the fully open hairpin was 174 bp, consistent with the actual DNA hairpin size (174 bp). To determine if pausing is related to DNA base-pair energy, we compared the unwinding dwell time histogram

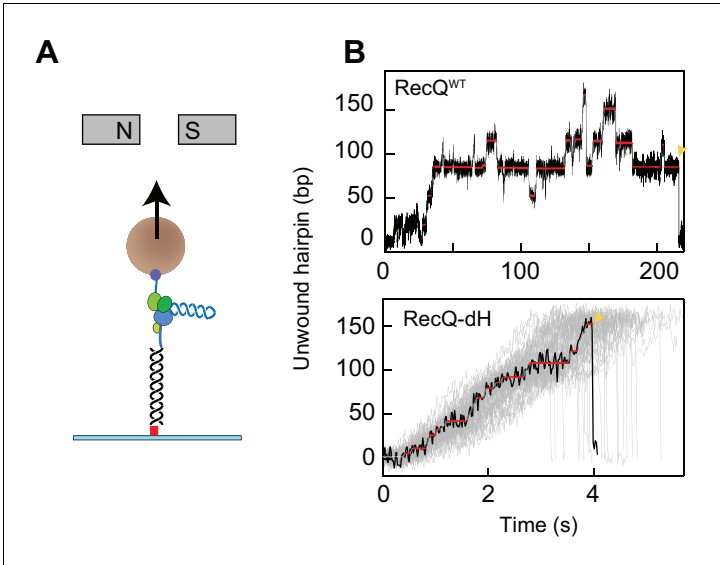

**Figure 1.** DNA hairpin unwinding activity of RecQ helicase is modulated by the HRDC domain. (**A**) Cartoon representation of the experimental scheme (not to scale). The 3′ biotinylated end of the single-stranded poly-dT segment (blue) is attached to a streptavidin-coated 1- or 2.8 μm magnetic bead (brown sphere), whereas the 5′ digoxigenin-labeled double-stranded handle (black line) is attached via anti-digoxigenin (red square) to the surface of the flow-cell. Small magnets above the flow-cell apply a constant upwards force on the magnetic bead. RecQ (purple and green RecA- like domains, yellow zinc binding and winged helix domain, orange HRDC domain) binds at the base of the hairpin (blue helix) and unwinds it, which results in the increase in the extension of the bead. (**B**) Individual unwinding events of RecQ$^{WT}$ and RecQ-dH. Unwound DNA indicates the amount of DNA hairpin opened by RecQ in base pairs. The ends of unwinding events are indicated by a yellow pointer. Pause locations identified from *T-test* fitting are indicated as solid red lines. Additional RecQ-dH unwinding traces are displayed to show the range of average unwinding rates (gray lines; note that only the region from the beginning to the maximum unwound positions are plotted).

DOI: https://doi.org/10.7554/eLife.45909.003

The following source data is available for figure 1:

**Source data 1.** Source data for *Figure 1*.
DOI: https://doi.org/10.7554/eLife.45909.004

(*Figure 2A*; top) with the DNA base-pair stability calculated by performing a running average (6 bp window) of the exponential of the DNA base-pair energy for the 174 bp DNA hairpin sequence based on the nearest neighbor base-pair energy model (*Patten et al., 1984*; *SantaLucia, 1998*; *Huguet et al., 2010*). We found that the peak locations of pausing and duplex stability were highly correlated (*Figure 2A*; bottom). The exact locations of peaks were identified by globally fitting the dwell-time histogram and the exponential of the average DNA melting energy with the sums of Gaussian distributions (*Figure 2—figure supplement 2*). The relationship between pausing during unwinding and the peaks in the dwell-time histogram is explained in *Figure 2—figure supplement 2B* (top). Consistent with this observation, pause positions from the dwell time histogram of RecQ$^{WT}$ were linearly correlated with peak positions from the DNA base-pair energy profile (*Figure 2B*; top) with a slope of 0.99 ± 0.03, linear correlation coefficient (Pearson's r) of 0.97, and $\chi^2 = 0.85$, indicating a strong linear correlation. The sequence around the peak positions (±4 bp) contained a high percentage of GC (~70%), consistent with the finding that the pause positions are related to the duplex stability of the DNA. This finding raises the question of how HRDC-dependent pausing is correlated with DNA base-pair melting despite the fact that the HRDC itself does not play a role in unwinding DNA or exhibit sequence-specific ssDNA binding. We hypothesized that the HRDC may amplify or stabilize transient pauses associated with RecQ core domain (RecQ-dH) encountering regions of increased duplex stability (high GC content).

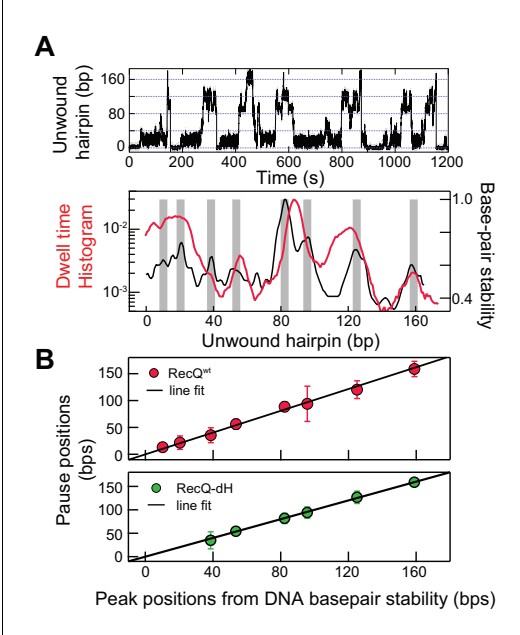

**Figure 2.** Sequence-dependent pausing of RecQ helicase. (**A**) An example trace (top), and dwell-time histogram of RecQ[WT] 174 bp hairpin unwinding trajectories (red line) plotted with the exponential of the DNA base-pair melting energy averaged over a 6 bp running window (black line) of the 174 bp DNA hairpin (bottom). The gray bars correspond to peaks in the dwell time histogram associated with RecQ pausing. (**B**) Linear regression analysis of pause positions of RecQ[wt] (top) and of RecQ-dH (bottom) plotted as function of the peak positions of DNA base-pair stability. The pause positions plotted as a function of energy peak positions were fit with straight lines returning fit values: slope of $0.99 \pm 0.03$, linear correlation coefficient (Pearson's r) of 0.97, and $\chi^2 = 0.85$ (top); a slope of $0.96 \pm 0.06$, Pearson's r = 0.95, and $\chi^2 = 1.1$ (bottom). The error bars correspond to the standard error of the mean (SEM).
DOI: https://doi.org/10.7554/eLife.45909.005

The following source data and figure supplements are available for figure 2:

**Source data 1.** Source data for *Figure 2—figure supplements 1* and *2*.
DOI: https://doi.org/10.7554/eLife.45909.008

**Figure supplement 1.** Sequence-dependent pauses are stabilized by the HRDC in RecQ[WT].
DOI: https://doi.org/10.7554/eLife.45909.006

**Figure supplement 2.** Multi-Gaussian peak fitting of calculated relative pause probability and dwell-time histogram of unwinding traces of RecQ[wt].
DOI: https://doi.org/10.7554/eLife.45909.007

## Sequence-dependent pausing originates from sequence-dependent unwinding kinetics

To test this hypothesis, we determined if the transient pausing positions of RecQ-dH correlated with the peaks in the DNA base-pair stability curve (*Figure 2B*). The pause positions for RecQ-dH were obtained from dwell time histograms following the same procedure used for RecQ[WT] (*Figure 2—figure supplement 2C*) and plotted as a function of the peak positions of DNA base-pair stability (*Figure 2B*). The pause positions of RecQ-dH were linearly correlated with the duplex stability peaks, returning a slope of $0.96 \pm 0.06$, Pearson's r = 0.95, and $\chi^2 = 1.1$. Moreover, the pause positions of RecQ-dH are statistically identical to those of RecQ[WT], confirming that HRDC-dependent pausing likely originates from stabilization of sequence-dependent unwinding kinetics of RecQ helicase. Sequence-dependent pausing by RecQ-dH reveals important mechanistic insights into the unwinding and translocation mechanism. If the enzyme unwinds one base pair per each kinetic step, the largest energy difference for a single base-pair opening (G/C vs A/T) is ~2.0 $k_BT$ so the pause duration ratio of G/C to A/T will be a maximum of ~7 fold. However, the roughly 20-fold difference in the time the enzyme requires to unwind DNA at the longest pause duration sites in comparison to the average unwinding rate, suggests that more than a single base pair is being opened by the enzyme during each kinetic step. Following this simple analysis, we suggest that pausing is governed by a combination of the DNA base-pair stability and the number of base pairs melted by the helicase during each kinetic step. During processive unwinding, this melting step is the rate limiting step that determines the unwinding rate and pause durations.

## Simulation of unwinding mechanism of RecQ reveals multi-base pair kinetic step

To distinguish among possible models for the unwinding mechanism of RecQ-dH based on its pausing behavior, we simulated unwinding trajectories comprising a series of pauses and translocations (*Figure 3A*).

Based on previous studies of helicases (*Manosas et al., 2010*; *Cheng et al., 2007*; *Neuman et al., 2005*; *Cheng et al., 2011*; *Lin et al., 2017*; *Myong et al., 2007*), we considered two scenarios for RecQ-dH unwinding with an *n*-bp kinetic step size: either the enzyme unwinds *n* base-pairs simultaneously then rapidly translocates along the unwound DNA (simultaneous melting

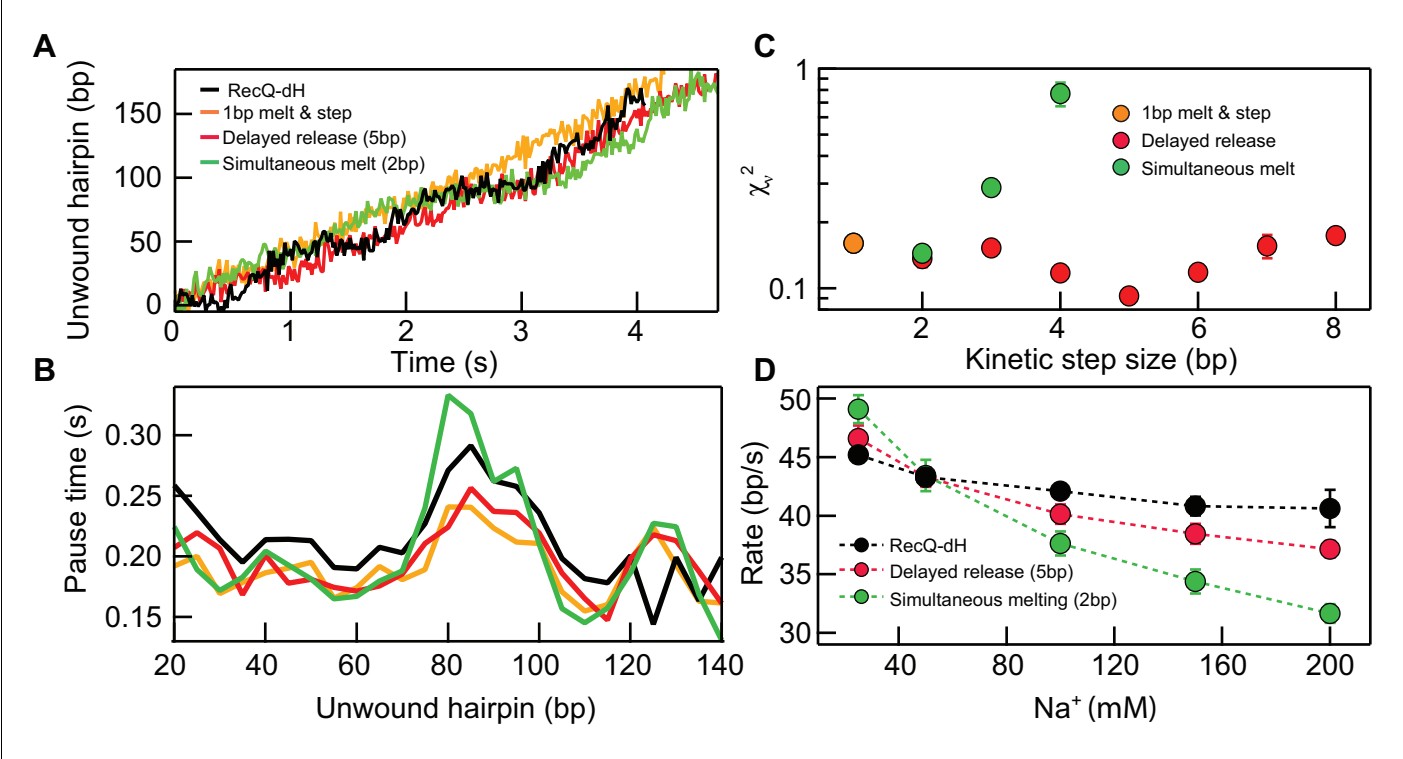

**Figure 3.** Kinetic modeling of the kinetic step-size. (A) Example simulated 174 bp DNA hairpin unwinding traces (1 bp melt and step (orange line), 1 bp melt and 5 bp step (red line), and 2 bp melt and step (green line)) overlaid with an example RecQ-dH unwinding trace (black line). Unwinding traces were simulated using *Equations 1 and 2*. The overall unwinding events are composed of pauses of lifetime ($\tau_p$) due to melting of the base-pairs, followed by a rapid translocation step in time ($\tau_t$). $\tau_p$ was calculated based on the sequence stability using nearest neighbor energy parameters (*Patten et al., 1984*; *SantaLucia, 1998*; *Huguet et al., 2010*). The total duration was adjusted to match the mean unwinding rate of RecQ-dH. (B) Pause times plotted as a function of the unwound hairpin for the three example models (with the same marker and line colors) in panel (A). Pause times and positions were obtained by analyzing simulated unwinding traces (100 traces for each condition) using *T*-test analysis and averaging pause times over a 5 bp window. The experimental pause lifetimes of RecQ-dH are shown in the black solid line. (C) Reduced $\chi^2$ ($\chi_v^2$) measure of the correspondence between measured and simulated pause durations as a function of pause position plotted as a function of the kinetic step size for three kinetic stepping models (see main text): 1 bp melt and step (orange filled circles), 1 bp melt and *n* bp step (red filled circles), and *n* bp melt and step (green filled circles). $\chi_v^2$ for 1 bp melt and step is significantly larger than the minima of the other two models. The $\chi_v^2$ is minimized for *n* = 2 bp for the *n*-bp melt and step model whereas $\chi_v^2$ is minimized for *n* = 5 bp for the 1 bp melt and *n* bp step model. (D) Na$^+$ dependent unwinding rates of RecQ-dH (black filled circles and dashed line) and predictions of the two kinetic models with the kinetic step-size, *n*, that minimizes $\chi_v^2$ for each model: 1 bp melt and 5 bp step (red filled circles and dashed line), and *2* bp melt and step (green filled circles and dashed line). The error bars correspond to the standard error of the mean (SEM).

DOI: https://doi.org/10.7554/eLife.45909.009

The following source data and figure supplement are available for figure 3:

**Source data 1.** Source data for *Figure 3* and *Figure 3—figure supplement 1*.
DOI: https://doi.org/10.7554/eLife.45909.011
**Figure supplement 1.** Comparison of simulated and experimental traces.
DOI: https://doi.org/10.7554/eLife.45909.010

model), or it sequentially unwinds *n* base-pairs then releases the newly melted ssDNA (delayed release model) (*Figure 3A*). We exclusively simulated RecQ-dH unwinding and pausing kinetics rather than RecQ$^{WT}$ due to the significantly more complex behavior of the RecQ$^{WT}$ unwinding trajectories (*Figure 1B*).

In the simultaneous melting model, the pause duration, $\tau$ is related to the sum of *n* base-pair energies at the position of the $i^{th}$ kinetic step,

$$\tau_p(i) = A_{RecQ}\exp\left(\sum_{s=1}^{n} G_{1bp}((i-1)+s)\right) \qquad (1)$$

Here $G_{1bp}$ is the free energy required for 1 base-pair melting at the $i^{th}$ position calculated using the nearest-neighbor energy parameters (*Patten et al., 1984*; *SantaLucia, 1998*; *Huguet et al., 2010*), $s$ is a step index ranging from 1 to $n$, and $A_{RecQ}$ is a pre-factor used to adjust the simulation to give the same average unwinding (46 nt/s) and translocation rate (~100 nt/s) as the RecQ-dH construct (*Manosas et al., 2010*). In the delayed release model, $\tau_p(i)$ is the sum of the pause times associated with melting each of $n$ base pairs at the $i^{th}$ kinetic step,

$$\tau_p(i) = \sum_{s=1}^{n} A_{RecQ} \exp\big(G_{1bp}((i-1)+s)\big) \qquad (2)$$

Stochastic simulations of both models were run with different step-sizes, $n$. For each value of $n$, the pre-factor $A_{RecQ}$ was adjusted to match the measured average rate of RecQ-dH, and the single-strand DNA translocation rate was 100 bp/s (*Manosas et al., 2010*; *Bagchi et al., 2018*). Simulated unwinding traces were generated for different kinetic step-sizes for the two different models (100 traces per each condition, example traces are shown in *Figure 3—figure supplement 1*). Simulated traces were analyzed with a *T*-test based step finding algorithm with the same parameters used for experimental data analysis. Pause durations were binned over 5 bp intervals for simulation and experimental traces and the mean pause duration for each bin was calculated (*Figure 3B*). Simulation results were compared with experimental data for RecQ-dH by calculating the reduced $\chi^2$ ($\chi_v^2$) between the simulated and experimental traces (*Figure 3C*). For the delayed release model, $\chi_v^2$ reached a minimum around 5 bp ($\chi_v^2 = 0.9 \times 10^{-1}$), lower than the minimum for simultaneous melting model that reached a minimum at 2 bp ($\chi_v^2 = 1.4 \times 10^{-1}$). This suggests that a delayed release scenario may describe the unwinding mechanism of core RecQ. To confirm this finding, we investigated how the RecQ-dH unwinding rate was affected by $Na^+$ concentration and compared the results with the two unwinding models. As DNA base-pair melting energy increases with $Na^+$ concentration (*SantaLucia, 1998*; *Huguet et al., 2010*), the average unwinding rate predicted by the simultaneous melting model should decrease more rapidly than that predicted by the delayed release model (*Figure 3D*). We varied the $Na^+$ concentration from 25 to 500 mM while maintaining $Mg^{2+}$ at 5 mM under otherwise identical buffer conditions. The unwinding rate of RecQ-dH decreased with increasing $Na^+$ concentration. The relative decrease in unwinding rate was much better described by the delayed release model with a 5 bp kinetic step. than the simultaneous 2 bp DNA melting model (*Figure 3D*). The small deviation between the delayed release model and the measured $Na^+$ concentration dependence of the unwinding rate suggests that although the duplex unwinding remains the rate-limiting step, the $Na^+$ concentration effects other aspects of unwinding such as protein-DNA interactions, which are beyond the scope of the simple model. Thus, to further test and confirm the delayed release unwinding model, we performed two additional experiments as explained below.

## Unwinding kinetics of forked DNA substrates in ensemble rapid kinetic experiments support the delayed release model

The significant DNA sequence dependence of the RecQ-catalyzed DNA unwinding rate and pausing characteristics detected in MT single-molecule experiments should be reflected in ensemble unwinding kinetic measurements, which are suitable for the determination of the kinetic step size and the macroscopic dsDNA unwinding rate (*Lucius et al., 2003*). In these experiments unwinding kinetics are monitored via the appearance of fully unwound reaction products. Thus, ensemble unwinding experiments complement MT experiments, in which individual unwinding steps are monitored. Importantly, these techniques together should allow determination of the microscopic unwinding mechanism of RecQ helicase constructs based on the proposed base-pair energy dependent unwinding models.

To test this idea, we performed single-turnover unwinding kinetic experiments in which we rapidly mixed complexes of RecQ$^{WT}$ or RecQ-dH with forked DNA substrates of varying GC content with ATP and excess unlabeled ssDNA traps in a quenched-flow instrument and monitored the time course of fluorescently-labeled ssDNA generation via gel electrophoresis of reaction products (*Figure 4A*). Forked DNA substrates used in the experiments comprised two 21-nt ssDNA arms and a 33 bp dsDNA segment containing 12 (gc36), 16 (gc48) or 26 GC (gc79) bps (sequences described in *Supplementary file 1*Table S1). Unwinding traces comprised a short (~0.1 s) initial lag, followed by a biphasic appearance of the labeled ssDNA reaction product (*Figure 4B*). The rapid rise

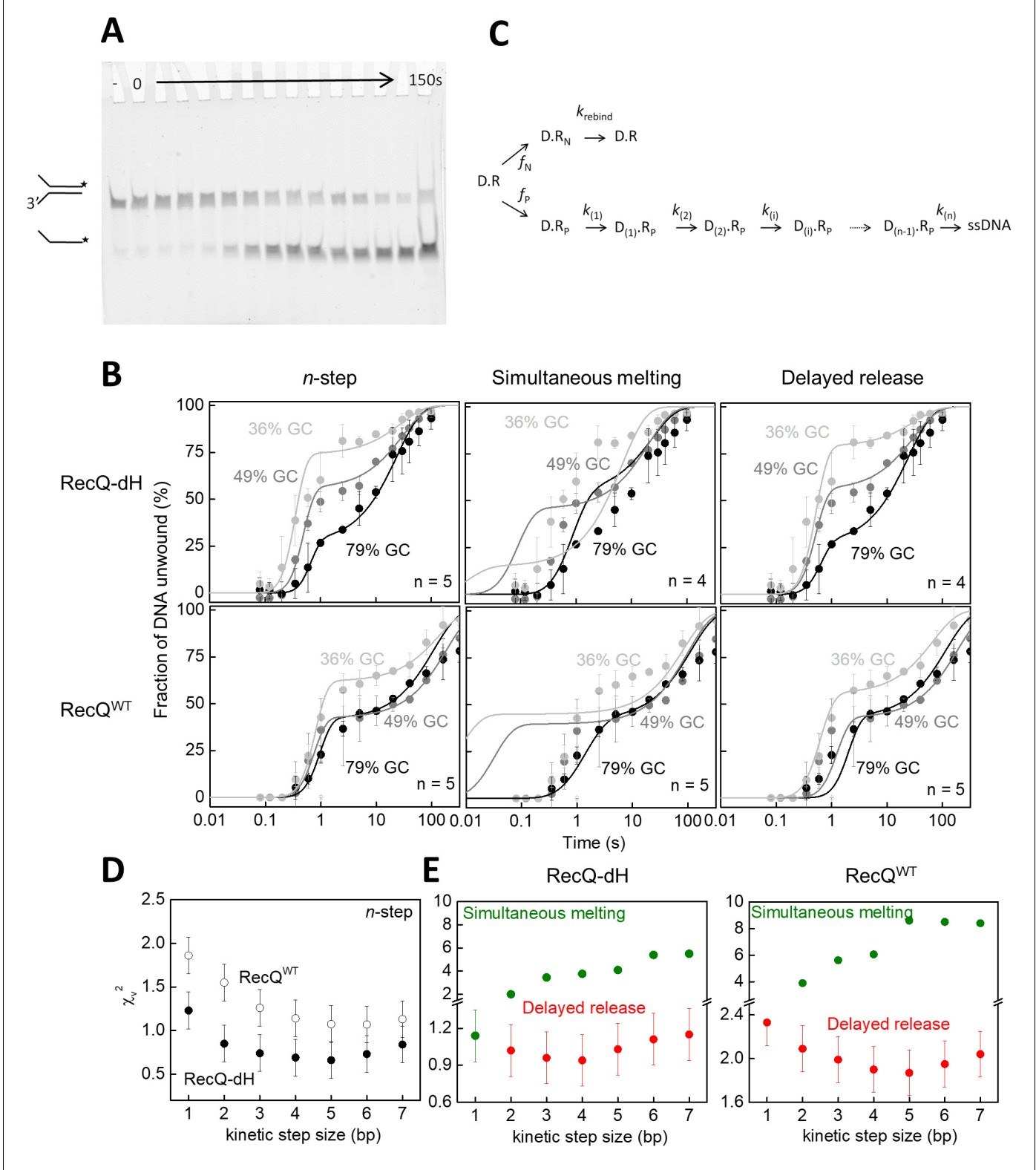

**Figure 4.** Single-turnover ensemble kinetic experiments. (A) Electrophoretogram of a single-turnover unwinding experiment. Preincubation of fluorescein-labeled forked duplex DNA (30 nM, gc36) with RecQ-dH (100 nM) was followed by rapid mixing with ATP (3 mM) plus excess ssDNA trap strand (3 μM) (final post-mixing concentrations). Reactions were stopped by the addition of EDTA (40 mM) and loading dye at different time points (0–150 s, cf. panel B) using a quenched-flow instrument or by manual mixing. Amounts of DNA species (forked duplex and ssDNA, depicted by cartoons) labeled with fluorescein (asterisk) were detected by a fluorescence imager. "– "denotes a 150 s control reaction in which ATP was absent. (B) Single-

*Figure 4 continued on next page*

*Figure 4 continued*

turnover unwinding kinetics of forked DNA substrates with GC contents of 36% (light gray), 48% (gray) and 79% (black) of RecQ-dH and RecQ$^{WT}$. Error bars represent SEM calculated from three experiments. Solid lines show fits based on: the *n*-step model at *n* = 5 for both helicase constructs (see scheme on panel C); simultaneous model (*Equation 1* and panel C) at *n* = 4 for RecQ-dH and *n* = 5 for RecQ$^{WT}$; and delayed release model (Equation. and panel C) at *n* = 4 for RecQ-dH and *n* = 5 for RecQ$^{WT}$. (C) Common scheme for the modified *n*-step and derived simultaneous melting and delayed release models. In the models, unwinding starts from the ssDNA-dsDNA junction. Of all DNA-RecQ complexes (D.R), only a fraction ($f_P$, D.R$_p$, lower row) unwinds the dsDNA segment in a single run, consisting of *n* consecutive irreversible kinetic steps ($k_{(1)} \ldots k_{(n)}$). In the *n*-step model, the rate constant of each unwinding kinetic step is identical ($k_{(1)} = k_{(n)}$). In contrast to this, in the simultaneous melting and delayed release models, the rate constant of each unwinding kinetic step depends on the base pair energy of the dsDNA segment to be unwound (segment length according to kinetic step size), as described in *Equation 1 and 2*, respectively. A fraction of DNA bound helicase molecules ($f_N$, D.R$_N$, upper row) is unable to successfully unwind DNA due to the limited processivity of the enzyme and/or more complex unwinding patterns. After dissociating from the DNA substrate, these enzyme molecules can rebind to the substrate at rate constant $k_{rebind}$ and start a new unwinding run (D.R). (D–E) Determined $\chi_v^2$ values from fitting the (D) *n*-step model for RecQ-dH (filled circles) and RecQ$^{WT}$ (open circles) or (E) fitting the simultaneous melting (green) and delayed release models (red) for the indicated helicase construct. Other determined parameters are listed in *Supplementary file 1* Table S2.

DOI: https://doi.org/10.7554/eLife.45909.012

The following source data is available for figure 4:

**Source data 1.** Source data for *Figure 4*.
DOI: https://doi.org/10.7554/eLife.45909.013

originated from single unwinding runs of initially DNA-bound helicase molecules. The slow rise phase originates from premature dissociation, followed by slow rebinding, of the enzyme to the DNA substrate (hindered but not totally inhibited by the ssDNA trap strand) that eventually led to full unwinding of the DNA fork (*Harami et al., 2017*). To obtain parameters of unwinding, data were analyzed with a modified version of a previously described *n*-step kinetic model (*Lucius et al., 2003*). In its simplest form the model assumes that DNA unwinding occurs as a result of *n* consecutive rate limiting steps that have a uniform rate constant. This model is generally suitable for the determination of the macroscopic dsDNA unwinding rate, the kinetic step size and the number of intermediates in the unwinding reaction (*Lucius et al., 2003*).

Using a modified version of the *n*-step model (*Figure 4C*), global fitting of the unwinding kinetics of the gc36, gc48 and gc79 substrates using an integer series of *n* ranging from 1 to 7 revealed smallest $\chi_v^2$ values for an apparent kinetic step size of 5 bp for both RecQ and RecQ-dH (*Figure 4B–D*) with all DNA substrates, similar to that suggested by our MT results (*Figure 3A–B*) and by previous findings (*Lin et al., 2017*; *Harami et al., 2015*). However, the *n*-step model does not consider the sequence dependence of the rates of elementary unwinding steps, precluding the distinction between different microscopic mechanisms producing the same kinetic step size. Therefore, we used the same physical framework as described for the MT experiments (*Equations 1 and 2*) and performed global kinetic fitting to all transients of a given helicase construct (RecQ$^{WT}$ or RecQ-dH) unwinding the different forked DNA substrates, based on the DNA sequence-dependent simultaneous melting and delayed release unwinding models (*Figure 4B–C*). For both models, fitting was done using an integer series of *n* ranging from 1 to 7. In agreement with the results of the MT analysis (*Figure 3C*), the smallest $\chi_v^2$ value was obtained for the delayed release model with a kinetic step size of 4 bp for RecQ-dH and 5 bp for RecQ$^{WT}$ (*Figure 4B and C*, other parameters are listed in *Supplementary file 1* Table S2).

## Direct measurement of 5 bp kinetic step size and time-dependent release of ssDNA

If RecQ-dH takes a certain kinetic step size, it could in principle be directly observed in the single-molecule unwinding traces. However, the enzyme unwinds DNA too rapidly at high ATP concentrations for steps to be routinely and accurately detected, given the spatial resolution limits of the measurement. Under our experimental conditions, the average baseline noise was ~14 nm at 200 Hz data collection rate. Thus, in order to observe, for example, a 4 bp step (i.e. a 3 nm change in DNA extension), the average pause duration should be >300 ms or the unwinding rate should be less than 13 bp/s (~3 fold slower than 42 bp/s). We tried three different conditions to decrease the unwinding rate of RecQ: lowering the ATP concentration (*Figure 5—figure supplement 1*) and including non-hydrolysable ATP analogues, ATPγS or AMP-PNP, in the assay (*Figure 5—figure*

*supplement 2*). We found that decreasing the ATP concentration (sufficiently lowering the unwinding rate) resulted in frequent and extensive enzyme backsliding (observable as rapid partial rezipping of the hairpin during an unwinding event), which complicates kinetic step size measurements (*Figure 5—figure supplement 1*). AMP-PNP showed extremely slow dissociation kinetics from RecQ-dH that were inappropriate for unwinding assays (*Figure 5—figure supplement 2*). On the other hand, ATPγS, showed a comparable binding affinity to ATP with a significantly shorter binding time (~1 s) than AMP-PNP (*Figure 5—figure supplement 2*). In addition, ATPγS binding transiently locks RecQ in the strong DNA-binding ATP bound state without backsliding, leading to long duration pauses that effectively increased the spatial resolution by permitting longer averaging times (*Figure 5A*). We measured the unwinding activity of RecQ-dH at different fractions of ATPγS (0.05– 0.5 mM) while keeping the total combined concentration of ATP and ATPγS constant at 1 mM. The unwinding rate decreased with increasing ATPγS fraction (*Figure 5A*). We reason that when the concentration of ATPγS is such that it is bound at least once per kinetic step, then the predominant

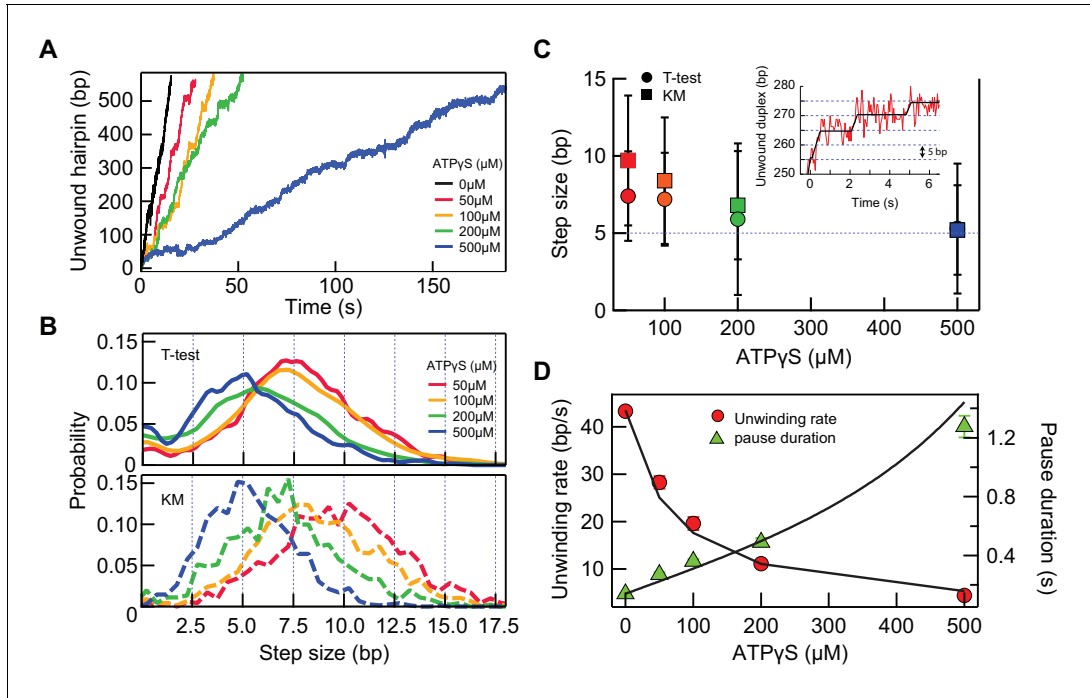

**Figure 5.** 5 bp kinetic step-size and tight mechano-chemical coupling. (**A**) Example traces of hairpin unwinding by RecQ-dH with increasing concentration of ATPγS while maintaining the combined concentration of ATP and ATPγS at 1 mM. (**B**) Step-size distributions were obtained by analyzing unwinding traces collected at each ATPγS concentration with either *T-test* or Kerssemakers (KM) step analysis algorithms (see main text). (**C**) The mean step-size obtained by fitting the distributions in panel (**B**) with Gaussians plotted as a function of ATPγS concentration. The average step sizes from both the *T-test* and Kerssemakers (KM) analysis converge to 5 bp with increasing ATPγS concentration. Inset: An example trace with 5 bp steps (red line) and the *T-test* fit (blue line). (**D**) Global fitting of the mean pause duration (green solid triangles) and average unwinding rate (red solid circles) as a function of ATPγS concentration, using *Equations 4 and 5*, reveal a tight mechano-chemical coupling ratio of *C* = 1.0 ± 0.2 bp/ATP.
DOI: https://doi.org/10.7554/eLife.45909.014

The following source data and figure supplements are available for figure 5:

**Source data 1.** Source data for *Figure 5* and *Figure 5—figure supplements 1–4*.
DOI: https://doi.org/10.7554/eLife.45909.019

**Figure supplement 1.** ATP dependence of unwinding, stepping, and backsliding kinetics of RecQ-dH.
DOI: https://doi.org/10.7554/eLife.45909.015

**Figure supplement 2.** Comparison of RecQ-dH unwinding the 584 bp DNA hairpin in the presence of AMPPNP or ATPγS and ATP.
DOI: https://doi.org/10.7554/eLife.45909.016

**Figure supplement 3.** Example trace of RecQ-dH at 500 µM ATPγS showing a 2.5 bp step.
DOI: https://doi.org/10.7554/eLife.45909.017

**Figure supplement 4.** RecQ-dH inefficiently hydrolyzes ATPγS.
DOI: https://doi.org/10.7554/eLife.45909.018

physical step-size measured in the hairpin unwinding trajectories will correspond to the kinetic step-size. Step-sizes were estimated with two different step finding algorithms: a step finding program originally developed by Kerssemakers and coworkers (*Kerssemakers et al., 2006*) and the *T*-test based step finding analysis (*Seol et al., 2016*). To determine the average kinetic step-size for each condition, the estimated step-sizes were histogrammed and fit with Gaussian distributions (*Figure 5B and C*). We found that the estimated step size of RecQ-dH from both step-finding algorithms were comparable, converging from ~8 bp at a low ATPγS fraction to 5 bp at higher ATPγS fractions, suggesting that the average kinetic step size of RecQ-dH is 5 bp [*T*-test: 5.3 ± 0.1 (center); 3.0 ± 0.6 (Standard Deviation); Kerssemakers: 5.2 ± 0.1(center); 2.1 ± 0.1 (Standard Deviation), errors correspond to the standard deviations from Gaussian fitting]. The broad step-size distribution could reflect the stochastic nature of ssDNA release by RecQ. Also, it is likely that the two ssDNA strands are released asynchronously by RecQ. In line with this, we occasionally observed a 2.5 bp kinetic step at 500 μM ATPγS, and the step-size distribution at lower ATPγS fractions included peaks at 7.5, 10, and 12.5 bp, consistent with a fundamental step-size of 2.5 bp corresponding to the release of one ssDNA strand of 5 nt (*Figure 5—figure supplement 3*).

## Tight mechano-chemical coupling of ATP-dependent unwinding by RecQ

The prolonged pause state due to ATPγS binding instead of ATP at the cleft between two RecA domains of RecQ enabled us to probe the mechano-chemical coupling, *C,* of RecQ helicase that is a measure of the number of ATP hydrolyzed per kinetic step. For $C = m/n$ (*m* ATP hydrolysis per *n* kinetic step size), the average number of bound ATPγS, *l*, can be estimated based on the binomial probability distribution.

$$l = \sum_{i=0}^{m} i \frac{m!}{i(m-i)!} P^i (1-P)^{m-i} \qquad (2)$$

$$P = \frac{k_{\text{ATP}\gamma S}[\text{ATP}\gamma S]}{k_{\text{ATP}}[\text{ATP}] + k_{\text{ATP}\gamma S}[\text{ATP}\gamma S]} \qquad (3)$$

*P* is the probability of ATPγS binding per each cycle. $k_{\text{ATP}}$ is the ATP on-rate, $k_{\text{ATP}\gamma S}$ is the ATPγS on-rate, and [ATPγS] and [ATP] are the concentrations of ATPγS and ATP, respectively. The hydrolysis rate of ATPγS by RecQ in the presence of excess $dT_{45}$ was measured by monitoring thiophosphate production (*Saran et al., 2006*) and estimated to be < 0.2/s (Appendix 1 and *Figure 5—figure supplement 1*). This is significantly slower than the measured pause escape rate (2.8 ± 0.1/s) at 500 μM ATPγS, suggesting that $k^{\text{off}}_{\text{ATP}\gamma S}$ is much faster than the rate of ATPγS hydrolysis. Thus, we could simplify the mean pause duration per kinetic step, τ and the mean unwinding rate, *v* as,

$$\tau = l \Big/ k^{off}_{ATP\gamma S} + 1 \Big/ k_{step} \qquad (4)$$

$$v = \frac{n}{\tau} \qquad (5)$$

where $k_{\text{step}}$ is the mean kinetic stepping rate without ATPγS. We obtained the average pause durations for different fractions of ATPγS from 5 to 50 % and globally fitted the pause durations and the average unwinding rates as a function of ATPγS concentration with Eq. 4 and 5 respectively (*Figure 5D*). From this global fitting, we found that $C = 1.0 ± 0.2$ bp/ATP, $k_{\text{ATP}}/ k_{\text{ATP}\gamma S} = 1.2 ± 0.2$, and $1/k^{\text{off}}_{\text{ATP}\gamma S} = 0.4 ± 0.1$ s suggesting a tight mechano-chemical coupling in agreement with previous ensemble measurements (*Harami et al., 2015*; *Sarlós et al., 2012*). We note that rebinding of ATPγS was not taken into account for simplicity in *Equations 2 and 3*, which is reasonable as ATPγS concentration is lower than ATP except for 50% ATPγS, and because the relative on rate ($k_{\text{ATP}}$[ATP] vs $k_{\text{ATP}\gamma S}$[ATPγS]) of ATPγS is lower than that of ATP. To ensure that this simplification is reasonable, we simulated how many ATPγS molecules are bound instead of ATP per base pair based on the fitting parameter, $k_{\text{ATP}}/k_{\text{ATP}\gamma S} = 1.2$ at 50% ATPγS. We found that the average is less than one ATPγS per base pair at this condition indicating that repetitive ATPγS binding at the same site is rare.

## Multi-step HRDC dependent pausing kinetics results in a non-linear amplification of intrinsic sequence-dependent pausing

The sequence-dependent unwinding mechanism of RecQ consisting of a 5 bp kinetic step results in transient pauses that are further stabilized by the HRDC, which results in the long-lived sequence-dependent pausing of RecQ$^{WT}$ (*Figure 2*). In addition to the sequence dependence, HRDC-dependent pausing exhibits two interesting features: occasional repetitive rezipping and unwinding (shuttling) around the pause position and significantly prolonged pausing durations for certain pausing positions (*Figure 2* and *Figure 2—figure supplement 1*). It appears that HRDC-binding triggers this shuttling behavior in which 5–10 bp are repetitively unwound and rezipped at the relatively long-lived (>0.14 s) intrinsic pause positions. Shuttling activity repeats until RecQ passes the sequence-dependent roadblock. This complex shuttling behavior was significantly enhanced at those regions where long pauses of RecQ-dH are clustered, such as at 55, 90, and 120 bps (*Figure 2—figure supplement 1*) resulting in the apparent high dwell probabilities at these sites (*Figure 2A*). Since these positions also exhibit the highest base-pair stabilities (*Figure 2A*), the average dwell time of RecQ$^{WT}$ is strongly correlated with the base-pair stability. Indeed, the average dwell times for RecQ$^{WT}$ are highly non-linearly correlated with the base-pair stability (*Figure 6A*). In contrast, the dwell-times for RecQ-dH scale linearly and much less dramatically with the base-pair stability (*Figure 6A*), indicating that the HRDC-stabilized pausing can be described as a non-linear amplifier of the intrinsic sequence-dependent unwinding kinetics. We found that a simple kinetic competition model in which HRDC binding is in kinetic competition with the forward motion of the helicase (Appendix 1) cannot reproduce the dramatic changes in pause probability observed for RecQ$^{WT}$ hairpin unwinding (*Figure 6—figure supplement 1*). In line with this, the pausing duration distribution for RecQ$^{wt}$ is better described by a double rather than single exponential distribution (*Figure 6—figure supplement 1B*). Both pause lifetimes (1.8 ± 0.3 s and 0.4 ± 0.1 s) are longer than the average pause duration (0.14 ± 0.03 s, see SI) for the core RecQ (RecQ-dH), indicating that there are multiple HRDC-dependent pause states.

To test the proposal that base-pair stability-dependent RecQ$^{WT}$ pausing underlies a potential mechanism of homology sensing, we investigated the effect of introducing single mismatches at high probability pause sites. We tested hairpin substrates containing 1, 2, or three single mismatches: (*i*) a mismatch introduced at 90 bp, (*ii*) mismatches introduced at 90 bp and 104 bp and (*ii*) mismatches introduced at 90 bp, 104 bp, and 124 bp. All mismatches were generated by changing G to T on the displaced strand (detailed sequence information is in Appendix 1). We found that pausing of RecQ$^{WT}$ around the 90 bp unwound hairpin position was significantly reduced compared to intact 174 bp hairpin when a mismatch was present at 90 bp and additional mismatch at 124 bp further suppressed pausing around 120 bp (*Figure 6B* and *Figure 6—figure supplement 2*). The effect of mismatches on pausing of RecQ$^{WT}$ can be clearly demonstrated by comparing the dwell-time histograms of three DNA substrates (*Figure 6B*). The prominent peaks in the dwell time histogram of the intact hairpin DNA were diminished one by one with the introduction of mismatches at the corresponding positions confirming the correlation between pausing and homology (*Figure 6B* and *Figure 6—figure supplement 2*).

## Discussion

RecQ helicases are well-established as critical enzymes that contribute to genome stability through their multiple roles in DNA repair and genome integrity. Whereas the mechanistic basis of many of the specialized activities of RecQ helicases have been established, the mechanism through which RecQ helicases can distinguish legitimate from illegitimate homologous recombination intermediates has not been established. We previously proposed a mechanistic model in which the HRDC domain of RecQ orients the enzyme to preferentially disrupt the strand invasion (or D-loop structure) corresponding to the earliest HR intermediate. In this model, discrimination between legitimate and illegitimate HR was achieved by modulating the degree of HRDC-induced pausing through an unknown mechanism. Here we demonstrate that HRCD-induced pausing is sequence-dependent and establish the mechanistic basis for this behavior. The pronounced pausing of WT RecQ at GC rich sequences results from two amplification steps that convert the $\sim k_BT$ energy differences between GC and AT base pairs to a robust readout of sequence stability (*Figure 6A*). The first amplification step arises from the basic mechanochemistry of DNA unwinding that involves a ~ 5 bp kinetic step in which ~ 5

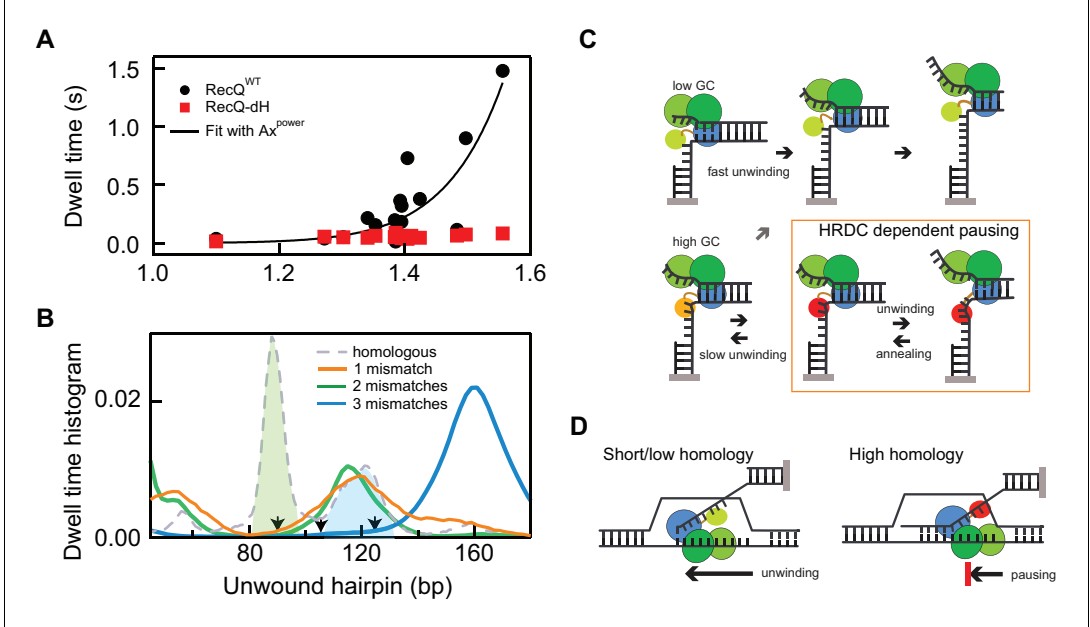

**Figure 6.** Non-linear amplification of sequence-dependent pausing by HRDC. (**A**) The average dwell times of RecQ$^{WT}$ and RecQ-dH plotted as a function of the exponential of the average base-pair energy over a 10 bp window. The non-linearity of RecQ$^{WT}$ dwell time data was analyzed with a power law function ($y = A \cdot x^{power}$ solid line). The fit returned A = 7 ± 0.9 (x$10^{-4}$) and Power = 17.2 ± 3.2 (Errors indicate standard deviations of fitting parameters). (**B**) Dwell-time histograms of RecQ unwinding the 174 bp DNA hairpin with zero mismatches, that is perfect homology, one mismatch (90 bp), two mismatches (90 and 104 bp), and three mismatches (90, 104, and 124 bp). The prominent peak around 90 bp (green shaded region) shown in the dwell time histogram of the intact DNA unwinding by RecQ$^{WT}$ was significantly reduced in the hairpin with a single mismatch and the additional peak around 120 bp (blue shaded region) was further suppressed by the third mismatch at 124 bp. The mismatch sites are indicated as arrows. (**C**) Model for D-loop homology discrimination via HRDC-mediated non-linear amplification of sequence-dependent pausing by RecQ. At regions of low GC content (upper row), RecQ rapidly unwinds duplex DNA and the HRDC remains in a weak binding ssDNA mode (light green HRDC). At regions of high GC content (lower row), RecQ pauses and the HRDC can switch to a strong ssDNA binding mode (orange HRDC). The subsequent binding of the HRDC to the displaced ssDNA (red HRDC) results in stabilization of the GC-induced pauses. As ssDNA is under tension or otherwise constrained, this interaction effectively hinders the movement of the RecQ core, resulting in short-range (5–10 bp) repetitive unwinding and annealing of DNA. (**D**) HRDC-dependent pausing regulates D-loop disruption in a homology dependent manner. RecQ can quickly unwind an invading strand of short or only partially homologous DNA, whereas HRDC-dependent pausing slows down unwinding and prevents disruption of an invading strand with an extended homology.

DOI: https://doi.org/10.7554/eLife.45909.020

The following source data and figure supplements are available for figure 6:

**Source data 1.** Source data for *Figure 6* and *Figure 6—figure supplements 1–2*.
DOI: https://doi.org/10.7554/eLife.45909.023

**Figure supplement 1.** HRDC-dependent pausing kinetics are inconsistent with a simple kinetic competition mechanism of pausing.
DOI: https://doi.org/10.7554/eLife.45909.021

**Figure supplement 2.** Example traces of intact (homologous) hairpin DNA, hairpin with one mismatch, hairpin with two mismatches, and hairpin with three mismatches by RecQ$^{WT}$.
DOI: https://doi.org/10.7554/eLife.45909.022

bp of DNA are unwound in 5 rounds of ATP hydrolysis followed by release of the two DNA strands. By coupling the forward motion of the helicase to the unwinding of 5 bp rather than a single bp, individual dwell times can vary ~5 fold more in relation to the underlying sequence than they would for single-bp steps. The second amplification step involves the non-linear amplification of the intrinsic difference in unwinding rate in proportion to GC content through the binding and stabilization of short sequence-dependent pauses by the HRDC. Together these two rounds of linear and non-linear amplification result in strong sequence dependent pausing of WT RecQ on hairpin substrates that results in a greater than 10-fold difference in average unwinding rate of legitimate versus illegitimate paired sequences. A recent single molecule study of *E. coli* RecQ$^{WT}$ indicated a switching behavior between a fast unwinding mode, similar to that of RecQ-dH, and a slower unwinding mode that is

similar but not identical to what we observed for RecQ$^{WT}$ unwinding in the present study (*Bagchi et al., 2018*). However, we did not observe switching of RecQ$^{WT}$ unwinding (frequent and prolonged pausing) to RecQ-dH-like unwinding (transient pausing). It is possible that the switching to the latter mode is caused by sequestration of the HRDC from ssDNA, which may be dependent on physicochemical conditions such as higher temperature and/or low salt concentrations.

Our study reveals how the unwinding mechanism of the core RecQ helicase, for example RecQ-dH, directly impacts HRDC-dependent pausing and the subsequent control of biological functions mediated by HRDC- dependent helicase activities. We were able to elucidate the coupling between ATP hydrolysis and the unusual kinetics of DNA unwinding by varying the ATP and ATPγS concentrations. We find that RecQ hydrolyzes 5 ATP molecules during a 5 bp kinetic unwinding step that concludes with asynchronous release of two five nt ssDNA segments on average (*Figures 3–5*). Furthermore, ATP binding likely stimulates RecQ binding to and melting of DNA duplex prior to hydrolysis (*Figure 5—figure supplement 1*).

## Sequence-dependent unwinding mechanism

The fundamental activity of helicases is the unwinding of duplex nucleic acids. In general, the unwinding mechanism has been classified as either passive or active depending on the degree to which the enzyme 'actively' destabilizes the duplex rather than 'passively' waiting for a thermal fluctuation to expose ssDNA (*Betterton and Jülicher, 2005*; *Lohman et al., 2008*). For a purely passive helicase, the enzyme does not provide external work to destabilize duplex DNA and translocates only when ssDNA is exposed by thermal fluctuations. On the other hand, an active helicase is actively involved in disrupting the DNA duplex, and in principle, is less sensitive to base-pair energy or sequence. In previous single molecule experiments, *E. coli* RecQ helicase was identified as an active helicase based on the minimal force and DNA sequence dependence of duplex unwinding (*Manosas et al., 2010*). In that study, following theoretical work by Betterton et al (*Betterton and Jülicher, 2005*), RecQ unwinding was modeled as one base-pair melting followed by 1–2 bases translocation. However, we found that the pause durations were generally longer than would be expected for melting of 1 base pair when we compared our results with simulations. We considered two different scenarios: RecQ either destabilizes multiple base-pairs ($\geq$2 bp) during each kinetic step similar to NS3 helicase (*Cheng et al., 2007*) or delays releasing of multiple unwound base-pairs similar to speculative models suggested in previous studies (*Cheng et al., 2011*; *Lin et al., 2017*; *Myong et al., 2007*; *Ma et al., 2018*). However, the minimal dependence of the unwinding rate on Na$^+$ concentration in addition to the sequence-dependent pauses cannot be explained by multi-base-pair melting. Rather, we found that an alternative scenario in which RecQ delays the release of nascent single-strand DNA (delayed release) was a better fit to the pause duration and Na$^+$-dependent unwinding rate data, though the associated kinetic step size (number of bp unwound prior to release) was not uniquely constrained by the pause duration or Na$^+$-dependent unwinding rate measurements (*Figure 3*). This finding is consistent with previous studies revealing 'asynchronous' release of nascent ssDNA (*Lin et al., 2017*; *Ma et al., 2018*). Nonetheless, the mechanism of delayed release of newly melted nucleotides remains unclear. Previous results suggest that a putative electrostatic interaction between newly melted ssDNA and RecQ sequesters several nucleotides of ssDNA. We consider a similar possibility in which RecQ releases the nascent ssDNA only when the accumulated torsion or tension on bound ssDNA is high enough to disrupt the interaction (*Myong et al., 2007*).

## RecQ takes 5-base kinetic steps and unwinds one base-pair per ATP hydrolysis

We further refined the delayed release model by directly measuring a 5 bp kinetic step size for DNA unwinding using ATPγS, which sufficiently slows down the unwinding rate without inducing the frequent back-sliding observed at reduced ATP concentrations (*Figure 5* and *Figure 5—figure supplement 1*)). Recent single molecule fluorescent studies showed 2–4 bp kinetic step (*Lin et al., 2017*; *Ma et al., 2018*). This smaller and more random nature of the kinetic step size is likely due to the low ATP concentration (2–5 μM), at which ATP binding likely becomes the dominant rate-limiting step slower than or on the same order as the intrinsic off-rate of the nascent DNA. Consistent with

this model, the study found a correlation between the ATP concentration and the measured kinetic step size.

The mechano-chemical coupling and kinetic analysis of ssDNA translocation of RecQ have been studied in detail (*Sarlós et al., 2012*; *Rad and Kowalczykowski, 2012*). Our unwinding kinetic step is consistent with a recent a study in which a five nucleotide kinetic step for RecQ translocation was estimated (*Rad and Kowalczykowski, 2012*). Other helicases display multi base-pair kinetic unwinding steps under sufficient ATP concentrations (above $K_M \sim 20~\mu M$) (*Lohman et al., 2008*). The mechano-chemical coupling is a measure of how many chemical cycles an enzyme completes to take one mechanical step. In the case of RecQ or other helicases, it corresponds to how many ATP molecules are consumed per one base translocation (or base-pair unwound for unwinding). For translocation, RecQ shows a tight coupling close to one nucleotide step per ATP hydrolysis (*Sarlós et al., 2012*). Our study reveals that the mechano-chemical coupling for unwinding maintains one base-pair melting per ATP hydrolysis (*Figure 4C*), which is also supported by the results of a recent single-molecule florescence study of RecQ unwinding (*Lin et al., 2017*). The proposed kinetic model based on our ATP dependent kinetic analysis (Appendix 1; *Figure 5—figure supplement 1*) suggests that DNA melting precedes ATP hydrolysis. In this model, ATP binding stabilizes the DNA-RecQ interaction and facilitates DNA melting presumably coupled to an ATP binding-dependent conformational change of RecQ such as rotation of the helicase domains relative to one another, which explains more frequent backsliding under lower ATP concentration (*Bernstein et al., 2003*; *Manthei et al., 2015*; *Pike et al., 2009*). Recent structural results suggest that RecQ binding may melt two base-pairs of DNA before ATP binding (*Manthei et al., 2015*). This may occur at the initial binding of RecQ (or rebinding) as the initiation of unwinding, but not the unwinding rate, is highly dependent on Na$^+$ concentration.

Whereas we establish that pausing arises from the stability of DNA duplex, recent work by Voter et. al suggests an alternative mechanism for sequence-dependent pausing. In their work, they identify a 'Guanine binding pocket' located in the helicase domain that specifically interacts with guanine bases to destabilize G-quadruplex structures. It is possible that these interactions could also slow down the unwinding rate at clusters of guanine bases in the translocation strand by inducing short pauses (*Voter et al., 2018*). However, the translocation sequence at the strong pause locations of our DNA hairpin is mixture of G and C bases, suggesting that the pauses we observed originate from the duplex stability. Nevertheless, we cannot entirely rule out the possibility that these specific guanine interactions contribute slightly to the pausing of RecQ core over and above the dominant effect of DNA duplex stability.

## HRDC amplifies weak sequence-dependent pauses during unwinding of RecQ core in a DNA substrate geometry-dependent manner

One of the essential aspects of RecQ is that it processes diverse, non-canonical, DNA substrates in which the HRDC plays an important role in modulating substrate-specific unwinding of RecQ. It has been shown that the HRDC regulates the binding orientation of RecQ core to promote disruption of D-loop structures, early homologous recombination intermediates (*Harami et al., 2017*). However, it was not clear how it can regulate unwinding of RecQ to selectively disrupt illegitimate or non-homologous invading DNA strands since the HRDC presumably cannot directly sense DNA sequence homology (*Harami et al., 2017*). Our present study reveals that the HRDC-ssDNA interactions are strongly coupled to DNA sequence-dependent pausing of the RecQ helicase core: ssDNA binding by the HRDC is not random but occurs at DNA sequences where the helicase core pauses due to the high duplex stability (*Figure 2*). On the other hand, either a low homology (base-pair mismatches) or low duplex stability (low GC regions) strongly reduces RecQ pausing (*Figures 2* and *6B* and *Figure 6—figure supplement 2*). Importantly, this feature can support discrimination between legitimate and illegitimate recombination events by RecQ helicases, in accordance with the increased illegitimate recombination frequencies detected in vivo upon compromising RecQ HRDC function (*Harami et al., 2017*; *Wang et al., 2016*). Recombination events proceed through the formation of a displacement loop (D-loop) flanked by genomic DNA, which, due to the limited mobility of these large DNA domains, mimics the hairpin geometry of the magnetic tweezers experiments in which the displaced DNA strand is constrained (*Figure 6C and D*). Previously we showed that the HRDC both targets RecQ to D-loop intermediates and orients the enzyme in a configuration favoring D-loop disruption (*Harami et al., 2017*). The results obtained here provide a mechanistic basis for

the subsequent discrimination between legitimate and illegitimate recombination based on the length and stability of the D-loop structure. RecQ-catalyzed unwinding of long and stable D-loops will be frequently interrupted by HRDC-stabilized pauses that drastically decrease the average unwinding rate. This slow average unwinding rate potentially permits the initiation of down-stream recombination processes associated with DNA synthesis resulting in extension and further stabilization of the D-loop. Conversely, RecQ unwinding of short and/or unstable D-loops will proceed rapidly (60–80 bp/s) resulting in the efficient disruption of the D-loop before it can be further extended. Our study reveals that the strategic location of the HRDC relative to the core domain, combined with sequence-dependent DNA unwinding, enable RecQ helicase to control pausing and shuttling in a substrate-dependent manner and expand its biological activity beyond simple duplex DNA unwinding. Whereas this study focused exclusively on *E. coli* RecQ, the homology sensing mechanism we propose is potentially applicable to the suppression of illegitimate, or so called 'homeologous recombination' by BLM (*Wang et al., 2016*).

Another biological role of HRDC domain-mediated pausing and shuttling could be linked to the role of RecQ helicases in G-quadruplex secondary DNA structure processing. G-quadruplex structures were shown to act as replication road blocks and these regions were shown to be recombinational hot spots (*van Wietmarschen et al., 2018*; *Rhodes and Lipps, 2015*). RecQ helicases can efficiently unwind G-quadruplex structures, possibly to aid DNA replication, suppress genome instability and to influence transcription of various genes (*Voter et al., 2018*; *Mendoza et al., 2016*). Prolonged shuttling at G-quadruplex sites could ensure that these secondary structures remain unfolded until further steps of replication or DNA repair can proceed. In line with this idea, the HRDC domain of human BLM helicase was shown to be essential for efficient, repetitive unwinding of G-quadruplexes *Chatterjee et al., 2014*).

In this study, we focused on elucidating the sequence-dependent unwinding and pausing mechanism of RecQ helicase in vitro with purified proteins. Whereas our results indicate a possible mechanism for homology sensing by RecQ helicases, the translocation and pausing kinetics on which the model is based could be modulated in vivo due to the interactions with other DNA binding and processing enzymes. For example, single-strand binding protein (SSB) would likely compete with the HRDC for ssDNA binding. However, we recently demonstrated that SSB is displaced by RecQ despite the much higher apparent binding affinity of SSB for ssDNA (*Mills et al., 2017*). Furthermore, the high local concentration of the HRDC, which is tethered to the RecQ core by a flexible linker, likely results in the HRDC out-competing other ssDNA binding proteins for the newly melted ssDNA. Nonetheless, as is often the case, RecQ helicases play diverse roles in DNA processing through the interaction with other proteins, thus, future experiments in the presence of other proteins that interact with RecQ in vivo including, SSB, RecJ, RecA, and Topoisomerase III are warranted to test our homology model in a context that more closely approximates physiological conditions.

# Materials and methods

## Key resources table

| Reagent type (species)or resource | Designation | Source/reference | Identifiers | Additional information |
|---|---|---|---|---|
| Strain, strain background (*Escherichia coli*) | ER2566 | New England Biolabs | NEB Cat. #: E4130 | |
| Recombinant DNA | Modified pTXB vector | PMID: 26067769 | | Transformation and expression of RecQ constructs |
| Recombinant DNA | pKZ1 | PMID: 28069956 | | Template for hairpin DNA substrate |
| Antibody | Anti-digoxigenin (Sheep Polyclonal) | Roche | Roche Cat# 11333089001, RRID:AB_514496 | Reconstituted in 1x Phosphate buffered saline (0.6 μg) |
| Commercial assay or kit | IMPACT purification system | New England Biolabs | NEB Cat. #: E6901S | |

*Continued on next page*

*Continued*

| Reagent type (species) or resource | Designation | Source/reference | Identifiers | Additional information |
|---|---|---|---|---|
| Commercial assay or kit | PCR DNA purification kit | Qiagen | Qiagen Cat. #: 28104 | |
| Chemical compound | Streptavidin coated magnetic beads (ø: 1 and 2.8 µm) | Invitrogen | Invitrogen Cat. #: 65602 and 65305 | |
| Chemical compound | Phusion high-fidelity DNA polymerase | New England Biolabs | NEB Cat. #: M0530 | |
| Chemical compound | T4 DNA ligase | Promega | Promega Cat. #: M1801 | |
| Chemical compound | Nt.BbvcI | New England Biolabs | NEB Cat. #: R0632 | |
| Chemical compound | BsaI-HF | New England Biolabs | NEB Cat. #: R3535 | |
| Software, algorithm | LabVIEW, Instrument control software | National Instruments | NI Cat. #: 776678–35 | |
| Software, algorithm | Igor pro 7, Data analysis | Wavemetrics PMID: 28069956 | | |
| Software, algorithm | MATLAB, Data analysis | MathWorks PMID: 16799566 | | |
| Software, algorithm | KinTek Global Kinetic Explorer 4.0, Data analysis | KinTek | | |

## DNA substrate preparation

### DNA hairpin substrates

Generation of the 174 bp DNA hairpin was previously described in detail (*Harami et al., 2017*). The 584 bp DNA hairpin was prepared by ligation of a 500 bp DNA hairpin with ~1.0 kb DNA handle. The 1.0 kb DNA handle was generated first by PCR of pKZ1, which contains two BbvcI sites spaced by 37 bp, between 4550 and 258 using one primer (258 position) containing a BsaI digestion site and the other primer (4550 position) labeled with 5′-digoxygenin. PCR products were digested by BsaI and gapped with Nt. BbvcI following the same method to generate the handle of the 174 bp DNA hairpin. 3′ biotin-labeled poly dT with a 33 bp region complementary to the gapped region of the 1 kb DNA handle was ligated to the 37-nt gapped region of 1 kb DNA handle. The 500 bp DNA hairpin was generated by PCR of Lambda DNA (NEB) between 23104 and 23608 and both ends were digested by BsaI. The final product was made by ligation of the 1 kb DNA handle with 3′ bio-tin-labeled poly dT, 500 bp DNA hairpin, and 12 bp DNA with a loop of 4 dT nucleotides to form the hairpin from the PCR product.

174 bp DNA hairpin with 1, 2, or 3 specific mismatch mutations (90 bp; 90 and 104 bps; 90, 104, 124 bps on the displacing strand) were generated by first cutting the PCR product for the 174 bp hairpin with NheI, yielding a 5′CTAG overhang. In order to prevent 100 bp fragments from NheI digestion to religate back to the DNA handle, the digested DNA band was extracted from an agarose gel. The two complementary oligos (88 nucleotides; Appendix 1) were annealed by incubation at 94°C for 5 min and then subsequent cooling to 4°C at a rate of −1 °C/s. The final product was made by ligation of the NheI-digested PCR product with 3′ biotin-labeled poly dT, 84 bp annealed DNA with differential four nt-overhangs, and 12 bp DNA with a loop of 4 dT nucleotides to form the hairpin.

## Enzyme preparation

The production of RecQ^WT and RecQ-dH were previously described in detail (*Seol et al., 2016*).

## Ensemble kinetic measurements

Forked DNA substrates (described in *Supplementary file 1* Table S1) were generated and single-turnover unwinding experiments were performed as in ref (*Harami et al., 2015*). Global fitting kinetic analysis was performed using KinTek Global Kinetic Explorer 4.0.

## Single-molecule measurements and data analysis

The magnetic tweezers and the experimental set-up were previously described (*Seol and Neuman, 2011*). A mixture of DNA hairpin (3 pmol) and anti-digoxigenin (0.5 μg) was incubated in phosphate buffered solution (PBS, pH 7.5) for 20 min and introduced into the sample chamber, which was incubated overnight at 4°C. The sample chamber was washed with 1 ml of wash buffer (WB, 1X PBS, 0.02 % v/v Tween-20, and 0.3 % w/v BSA) to remove unbound DNA molecules and 40 μl of magnetic beads (MyOne, Invitrogen) were introduced to form DNA hairpin tethers. Correct DNA hairpins were identified by the sharp DNA extension change upon DNA hairpin unfolding under high force (~19 pN). Upon finding a proper DNA substrate, the chamber was washed with 200 μl of RecQ buffer (30 mM Tris pH 8, 50 mM NaCl, 5 mM MgCl$_2$, 0.3 % w/v BSA, 0.04 % v/v Tween-20, 1 mM DTT, and 1 mM ATP). After washing, RecQ was added at a concentration of 20–100 pM in 200 μl RecQ buffer. DNA unwinding measurements were done by tracking a DNA tethered magnetic bead in real-time with custom written routines in Labview. During the measurement, a 1 μm polystyrene stuck bead was tracked to correct sample cell drift by adjusting the sample cell position using 3-D piezo stage (Physik Instrumente) to compensate for the drift. The unwinding traces were analyzed with a custom-written *T-test* based algorithm in Igor Pro 6 (Wavemetrics) and the Kerssemakers step finding program in MatLab (*Seol et al., 2016*; *Kerssemakers et al., 2006*; *Carter and Cross, 2005*).

## Acknowledgements

We thank Yasuharu Takagi, Sarah Heissler and Jim Sellers for assistance with ATP hydrolysis assays and Dr. Jonathan Silver for comments. This research was supported by the Human Frontiers Science Program (RGY0072/2010) and the Intramural Research Program of the National Heart, Lung, and Blood Institute, National Institutes of Health (to KCN); the 'Momentum' Program of the Hungarian Academy of Sciences (LP2011-006/2011), ELTE KMOP-4.2.1/B-10-2011-0002, NKFIH K-116072, NKFIH ERC_HU 117680, and NKFIH K-123989 grants (to MK); and a Premium Postdoctoral Fellowship of the Hungarian Academy of Sciences (to GMH).

## Additional information

### Funding

| Funder | Grant reference number | Author |
|---|---|---|
| Human Frontier Science Program | RGY0072/2010 | Yeonee Seol<br>Gábor M Harami<br>Mihály Kovács<br>Keir C Neuman |
| National Institutes of Health | HL001056-12 | Yeonee Seol<br>Keir C Neuman |
| Hungarian Academy of Sciences | LP2011-006/2011 | Gábor M Harami<br>Mihály Kovács |
| Eötvös Loránd University | ELTE KMOP-4.2.1/B-10-2011-0002 | Mihály Kovács |
| Nemzeti Kutatási és Technológiai Hivatal | NKFIH K-116072 | Mihály Kovács |
| Nemzeti Kutatási és Technológiai Hivatal | NKFIH ERC_HU 117680 | Mihály Kovács |
| Nemzeti Kutatási és Technológiai Hivatal | NKFIH K-123989 | Mihály Kovács |

The funders had no role in study design, data collection and interpretation, or the decision to submit the work for publication.

### Author contributions
Yeonee Seol, Conceptualization, Resources, Data curation, Software, Formal analysis, Validation, Investigation, Visualization, Methodology, Writing—original draft, Project administration, Writing—review and editing; Gábor M Harami, Conceptualization, Data curation, Software, Formal analysis, Investigation, Methodology, Writing—original draft, Writing—review and editing; Mihály Kovács, Conceptualization, Formal analysis, Supervision, Funding acquisition, Investigation, Methodology, Writing—original draft, Project administration, Writing—review and editing; Keir C Neuman, Conceptualization, Software, Formal analysis, Supervision, Funding acquisition, Validation, Investigation, Methodology, Writing—original draft, Project administration, Writing—review and editing

### Author ORCIDs
Mihály Kovács (iD) https://orcid.org/0000-0002-1200-4741
Keir C Neuman (iD) https://orcid.org/0000-0002-0863-5671

### Decision letter and Author response
Decision letter https://doi.org/10.7554/eLife.45909.029
Author response https://doi.org/10.7554/eLife.45909.030

## Additional files

### Supplementary files
• Supplementary file 1. Supplementary Tables including DNA sequences and fitting paramters. Table S1: Sequence (5' to 3') of ssDNA strands composing gc36, gc46 and gc79 forked dsDNA substrates (complementary regions are bold; Flu represents 3' fluorescein labeling). Table S2: Parameters determined from global fitting ensemble unwinding data with the $n$-step and delayed release models [a] Parameters determined from fitting ensemble unwinding data with the $n$-step model with $n = 5$ (cf. *Figure 4B*). [b] Parameters determined from fitting ensemble unwinding data with the delayed release model with $n = 5$ for RecQ$^{WT}$ and $n = 4$ for RecQ-dH (cf. *Figure 4B*). Table S3 Parameters from fitting of $P_{off}$, $\tau_{off}$, and $v$ as a function of ATP with three schemes. Table S4 Number of events in the analysis [a]: Number of events (Na$^+$ mM), [b]: Number of events (ATPγS μM), [c]: Number of events (ATP μM) $K_M k_a$
DOI: https://doi.org/10.7554/eLife.45909.024

• Transparent reporting form
DOI: https://doi.org/10.7554/eLife.45909.025

### Data availability
The single molecule experimental data analysis codes in this study were previously published and referenced in the manuscript. The Kerssemakers step-finder routine (Kerssemakers et. al. (2006) *Nature* 442:709-712) is available from the authors. Alternative step-finding routines (Wiggins (2015) *Biophys J* 109:346-354; Hill et al. (2018) *J Chem Phys* 148:123317) are available online (at http://mtshasta.phys.washington.edu/website/steppi/ or https://github.com/duderstadt-lab/Julia_KCP_Notebooks). Source data for all of the figures and graphs are provided in the main and supplemental data.

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

## Appendix 1

DOI: https://doi.org/10.7554/eLife.45909.026

### Kinetic analysis of ATP-dependent unwinding and backsliding reveals that DNA melting precedes ATP hydrolysis

Usage of limiting ATP concentrations in hairpin unwinding experiments revealed a kinetic state of RecQ-dH that branched off the unwinding pathway, which enabled us to formulate a kinetic scheme relating ATP binding and hydrolysis by RecQ to its unwinding and stepping activities.

We considered three potential schemes for the coupling between the ATP binding and hydrolysis cycle and enzyme translocation. Each scheme consists of a different arrangement of four steps: DNA melting, ATP binding, hydrolysis, and RecQ translocation along ssDNA. In scheme A, RecQ binding triggers DNA melting that allows ATP binding and hydrolysis, followed by translocation of RecQ. In scheme B, DNA melting and immediate translocation of RecQ are driven by ATP hydrolysis. In scheme C, DNA melting occurs after ATP binding followed by ATP hydrolysis and translocation of RecQ.

Upon decreasing the ATP concentration from 1 mM to 5 µM, we observed sudden decreases in DNA hairpin extension, corresponding to rapid re-zipping of the hairpin that terminated in a pause followed by the resumption of unwinding. The probability and duration of these backsliding and pausing events increased with decreasing ATP concentration, consistent with an off-pathway weakly DNA-bound RecQ state in kinetic competition with ATP binding. We obtained the off-pathway probability ($P_{off}$) by counting the number of sudden extension decrease events (**Figure 5—figure supplement 1A**) normalized by the number of unwound base-pairs for each trace. To obtain the off-state duration ($\tau_{off}$), we measured individual pause durations between sudden DNA extension decrease (re-zipping) events and the resumption of DNA unwinding (**Figure 5—figure supplement 1A**). For each kinetic scheme, $P_{off}$ can be calculated by summing $P_{off}$ for all possible pathways while the off-state duration ($\tau_{off}$) can be calculated using a recursive relation (**Shaevitz et al., 2005**).

As some of the kinetic rates are common in $P_{off}$, $\tau_{off}$, and the unwinding velocity, $v$, we globally fitted the experimentally measured $P_{off}$, $\tau_{off}$, and $v$ as a function of ATP concentration with the analytical expressions for $P_{off}$ and $\tau_{off}$ for each scheme and $v$ (see below) resulting in $k_{on}$, $k_{off}$, $K_M$ (=$k_{ra}/k_a$), $k_p$, $n$ and $k_{cat}$. The kinetic parameters are defined as follows:

| Parameter | Description |
| --- | --- |
| $D_n$ | DNA hairpin unwound by $n$ base pairs |
| $R$ | RecQ enzyme |
| $D+R$ | Weakly bound DNA-RecQ state |
| $D \bullet R$ | Tightly bound DNA-RecQ state, competent for unwinding |
| $D_n^*$ | DNA unwound by $n$ base pairs with one additional melted base-pair |
| $k_a$ | ATP binding rate to RecQ during DNA unwinding |
| $k_{ra}$ | ATP dissociation rate from RecQ during DNA unwinding |
| $k_h$ | ATP hydrolysis rate during DNA unwinding |
| $k_{on}$ | Rate from off-pathway weak DNA binding state to tight DNA binding state |
| $k_{off}$ | Rate from tight DNA binding state to weak, off-pathway, binding state |
| $K_M$ | ATP dissociation constant ($k_{ra}/k_a$) |
| $k_{cat}$ | Maximum unwinding rate of RecQ |
| $n$ | Coupling between unwinding rate and ATP hydrolysis rate |

*continued*

| Parameter | Description |
|---|---|
| $k_m$ | Rate of melting one DNA base-pair by RecQ |
| $k_p$ | Rate of reannealing one DNA base-pair by RecQ |
| $k_{fut}$ | Rate of RecQ failing to translocate |
| $k_{ss}$ | Rate of RecQ translocating a single base |

In order to decrease the free parameters, we set $k_a$, and $k_h$ and $k_m$ to be 2 (s$^{-1}$μM$^{-1}$), 200 (s$^{-1}$), and 121 (s$^{-1}$), respectively, based on previous measurements (*Sarlós et al., 2012*) and the average unwinding time from the experimental measurements and used $K_M k_a$ $\chi 2$ instead of $k_{ra}$ $\chi 2$ in the fitting. The Chi-squared value ($\chi^2$) from the global fitting with scheme A is the minimum among the three as $\chi^2$ values for schemes A, B, and C are 82, 604, and 9420, respectively. This suggests that ATP binding and DNA melting of RecQ may precede ATP hydrolysis, as encompassed in scheme **A**.

## Three kinetic schemes of RecQ unwinding and translocation for single-molecule measurement analysis

For all schemes, *R* is RecQ, *D* is DNA, *n* indicates the current DNA base-pair position where RecQ binds and performs its catalytic activity and $D_n$* represents a DNA state in which the *n +1* st DNA base-pair is melted prior to translocation, which results in $D_{n+1}$.

### A ATP binding and DNA melting

$$D + R$$
$$k_{off} \uparrow\downarrow k_{on}$$
$$D_n \cdot R \overset{k_a}{\underset{k_p}{\rightleftharpoons}} D_n \cdot R \cdot ATP \xrightarrow{k_n} D_n \cdot R \cdot ADP \cdot Pi \xrightarrow{k_n} D_{n+1} \cdot R$$
$$\Big\downarrow k_{fut}$$
$$D_n \cdot R$$

The probability of RecQ entering the weakly bound state (from $D_n \cdot R$ to $D_n + R$), $P_{off}$, and $\tau_{off}$ for scheme A are:

$$P_{off} = \frac{k_{off}}{k_{off} + k_m} \left[ 1 + \frac{k_m k_p (k_{ra} + k_h)}{(k_{off} + k_m) k_a [ATP] k_h + (k_{ra} + k_h) k_m k_p} \right]$$

$$\tau_{off} = \frac{1}{k_a [ATP]} + \frac{1}{k_{on}} \left( \frac{k_{off}}{k_a [ATP]} + 1 \right) + \frac{1}{k_h} + \frac{1}{k_m} \left[ 1 + \frac{k_p}{k_h} + \frac{k_{ra} (k_{on} + k_{off})(k_p + k_h)}{k_h k_{on} k_a [ATP]} \right]$$

### B DNA melting and ATP binding

$$D + R$$
$$k_{off} \uparrow\downarrow k_{on}$$
$$D_n \cdot R \overset{k_m}{\underset{k_p}{\rightleftharpoons}} D_n \cdot R \overset{k_a}{\underset{k_{ra}}{\rightleftharpoons}} D_n^* \cdot R \cdot ATP \xrightarrow{k_n} D_n^* \cdot R \cdot ADP \cdot Pi \xrightarrow{k_n} D_{n+1} \cdot R$$
$$\Big\downarrow k_{fut}$$
$$D_n \cdot R$$

$P_{off}$, and $\tau_{off}$ for scheme B are:

$$P_{off} = \frac{k_{off}}{k_{off} + k_a[ATP]} \left[ 1 + \frac{k_a[ATP]k_{ra}(k_p + k_h)}{(k_{off} + k_a[ATP])k_m k_h + (k_p + k_h)k_a[ATP]k_{ra}} \right]$$

$$\tau_{off} = \frac{1}{k_m} + \frac{1}{k_{on}}\left(\frac{k_{off}}{k_m} + 1\right) + \frac{1}{k_h} + \frac{1}{k_a[ATP]}\left[ 1 + \frac{k_{ra}}{k_h} + \frac{k_p(k_{on} + k_{off})(k_{ra} + k_h)}{k_h k_{on} k_m} \right]$$

## C ATP binding and hydrolysis follow by DNA melting

$$D + R$$
$$k_{off} \uparrow\downarrow k_{on}$$
$$D_n \cdot R \underset{k_{ra}}{\overset{k_a}{\rightleftharpoons}} D_n \cdot R \cdot ATP \overset{k_h}{\rightarrow} D_n \cdot R \cdot ADP \cdot Pi \underset{k_p}{\overset{k_m}{\rightleftharpoons}} D_n \cdot R \cdot ADP \cdot Pi \overset{k_{ss}}{\rightarrow} D_{n+1} \cdot R$$
$$\downarrow k_{fut}$$
$$D_n \cdot R$$

$P_{off}$ and $\tau_{off}$ for the scheme C are:

$$P_{off} = \frac{k_{off}(k_h + k_{ra})}{k_{off}k_h + k_{off}k_{ra} + k_a[ATP]k_h}$$

$$\tau_{off} = \frac{1}{k_{on}} + \frac{1}{k_h} + \frac{1}{k_a[ATP]}\left[\frac{(k_{ra} + k_h)(k_{off} + k_{on})}{k_{on}k_h}\right]$$

The dependence of the unwinding velocity, $v$, on ATP concentration with a Hill coefficient $r$:

$$v = \frac{k_{cat}}{1 + \left(\frac{K_M}{ATP}\right)^r}$$

## Adenosine 5′-O-(3-thiotriphosphate) (ATPγS) hydrolysis measurement

ATPγS hydrolysis by RecQ-dH was estimated by measuring the level of thiophosphate in the reaction mixture using Malachite Green (BioAssay System; POMG-25H). First, the thiophosphate standard was generated by measuring the OD at 620 nm of 8 different thiophosphate concentrations (0, 1, 2, 4, 8, 10, 20, and 40 µM) in RecQ activity buffer. Using this standard, the amount of thiophosphate production from ATPγS hydrolysis by RecQ was estimated by measuring OD at 620 nm of the reaction mixtures containing 200 nM dT$_{54}$ DNA, 1 mM ATPγS and 200 nM RecQ at 0, 30 min, 1, 2, and 3 hr reaction times. Concurrently, the OD of the same reaction mixture without RecQ was also measured to correct any effects of auto-hydrolysis of ATPγS.

## HRDC-dependent pausing analysis and simulation

In a simple kinetic scenario, HRDC-dependent pausing can be considered as an off-pathway state with a single rate-limiting step, $k_{HP}$ (the rate of entering an HRDC-stabilized pause state) that is in kinetic competition with the forward unwinding rate, $k_{step}$. The rate of escaping from the pause, that is the inverse lifetime of the HRDC-stabilized pause state or unbinding rate is $k_{-HP}$ (*Herbert et al., 2006*). As there are no potential long pauses before or after the peak at 40 bp (*Figure 2*), we can estimate the probability and kinetics of HRDC-dependent pausing related to the sequence-dependent pausing. Although the average pause duration at 40 bp for RecQ$^{WT}$ (0.16 ± 0.03 s) is comparable to that of RecQ-dH (0.14 ± 0.03 s), the long pauses (>0.5 s) were observed only in the traces of RecQ$^{WT}$ but not RecQ-dH hairpin unwinding suggesting that they represent HRDC binding (*Figure 6—figure*

*supplement 1*). At the 40 bp hairpin position, the step rate, $k_{step}$ (inverse of pause duration for RecQ-dH) is ~7 s$^{-1}$ and the HRDC-stabilized pause efficiency is 0.25, estimated from the fraction of the RecQ$^{WT}$ pause duration distribution that exceeds the exponential fit with $k_{step}$ (*Figure 6—figure supplement 1*). The estimated $k_{HP}$ is then 2.4 s$^{-1}$ and the estimated $k_{-HP}$ is 0.89 s$^{-1}$. Based on the sequence-dependent stepping rate $k_{step}$ and the calculated $k_{HP}$ and $k_{-HP}$ rates, HRDC-dependent pauses were calculated over the entire 174 bp hairpin. We found that the calculated pause durations and pause locations based on these simple assumptions fail to reproduce the RecQ$^{WT}$ pause behavior (*Figure 6—figure supplement 1*). In general, the pause locations are less localized, with HRDC-stabilized pauses occurring randomly throughout the hairpin sequence, and the average lifetimes of the specific pauses are shorter, in the simulations as compared to the RecQ$^{WT}$ data. This suggests that the pathway into and out-of the HRDC-stabilized pause may not be a simple single-step process, but rather a multi-phasic kinetic step. In support of this possibility, we found that the pausing duration distribution is better described by double-exponential ($\chi_v^2 = 2.1$) than single-exponential ($\chi_v^2 = 3.2$) (*Figure 6—figure supplement 1B*).

## DNA hairpin sequence information

The sequence information indicates the 3′ to 5′ translocating strand (green region) for all DNA hairpins.

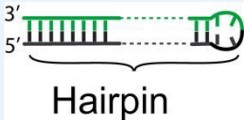

**Appendix 1—figure 1.** Cartoon of DNA haripin used for single-molecule measurements.
DOI: https://doi.org/10.7554/eLife.45909.027

### 174 base-pair DNA hairpin (translocation strand)

3′GTCGAAGGCTGACGTCGGACTGCGGTCCCGACTCCACTAGGGCTGGG
TAAACGACAGGTGGTCAGT
ACGATCGGTATACCGACGGCGCGCCGTGGTCCGGCGACGACACTACTACTACTAC
TACCGACGACGGGTACCATATAGAGGAAGAATTTCtggtgtaccgtcatcctt

### 174 base-pair DNA hairpin containing mismatches in the displacing strand

Two mismatches: G(90) →T; [Original sequence (position in the hairpin)→ mutated sequence]

 5′CAGCTTCCGACTGCAGCCTGACGCCAGGGCTGAGGTGATCCCGACCCATTTGCTGTCC
ACCAGTCATGCTAGCCATATGGCTGCCGCGCTGCACCAGGCCGCTGCTGTGATGATGA
TGATGATGGCTGCTGCCCATGGTATATCTCCTTCTTAAAGaccacatggcagtaggtt

 Two mismatches: G(90) →T; G(104)→T; [Original sequence (position in the hairpin)→ mutated sequence]

 5′CAGCTTCCGACTGCAGCCTGACGCCAGGGCTGAGGTGATCCCGACCCATTTGC
TGTCCACCAGTCATGCTAGCCATATGGCTGCCGCGCTGCACCAGGCCGCTTCTGTGAT
GATGATGATGATGGCTGCTGCCCATGGTATATCTCCTTCTTAAAGaccacatggcagtaggtt

 Three mismatches: G(90) →T; G(104)→T; G(124)→T; [Original sequence (position in the hairpin)→ mutated sequence]

 5′CAGCTTCCGACTGCAGCCTGACGCCAGGGCTGAGGTGATCCCGACCC
ATTTGCTGTCCACCAGTCATGCTAGCCATATGGCTGCCGCGCTGCACC
AGGCCGCTTCTGTGATGATGATGATGATTGCTGCTGCCCATGGTATATCTCCTTCTTAAAG
accacatggcagtaggtt

## Oligos that were used to generate 174 bp with mismatches

Translocation strand

 5´TGGTCTTTAAGAAGGAGATATACCATGGGCAGCAGCCATCATCATCATCATCACAGCAGCGGCCTGGTGCCGCGCGGCAGCCATATGG

 Displacing strand

 5´CTAGCCATAtGgCTGCCGCGCTGCACCAGGCCGCTGCTGTGATGATGATGATGATGGCTG

 CTGCCCATGGTATATCTCCTTCTTAAAG (one mismatch)

 5´CTAGCCATAtGgCTGCCGCGCTGCACCAGGCCGCTTCTGTGATGATGATGATGATGGCTG

 CTGCCCATGGTATATCTCCTTCTTAAAG (two mismatches)

 5´CTAGCCATAtGgCTGCCGCGCTGCACCAGGCCGCTTCTGTGATGATGATGATGATTGCTG

 CTGCCCATGGTATATCTCCTTCTTAAAG (three mismatches)

### 584 base-pair DNA hairpin

3´cagcttccgactgcagcctgacgccagggctgaggtgatccgcgacccatttgct
gtccaccagtcaacacgcaaaggctactcttatttcatcttacattgaagaaaatg
acattgagtttattacaaatgaaagtatgttaaacattggtataaaaaagttagtcagact
tgcacaaaagaaattaccttcatatttaacgaatcatctattattaaatcagaaagacgatg
ggtcgctaatacgctaaaagattacgaattattacatgccttagctataatctatggcagaat
gtataactgctgtaactctcttggcatacaaataaacaatccaatgggtgacgatgtgatttcgc
caacatcattcgactctttatttgatgaagccaggagaataacttatttaaaattaaaagattactccat
aagcaaattgtcatttagcatgatacaatatgacaataaaataattcctgaagatattaaagagcgtc
taaaactggtagataagcctaaaaatatcacttcgacagaagagttagttgactatacagccaagaccacatggcagtaggtt

