## [Decision Letter]

Thank you for submitting your article "Legitimate homology sensing via non-linear amplification of sequence dependent pausing by RecQ helicase" for consideration by *eLife*. Your article has been reviewed by three peer reviewers, one of whom is a member of our Board of Reviewing Editors, and the evaluation has been overseen by John Kuriyan as the Senior Editor. The following individual involved in review of your submission has agreed to reveal his identity: James Keck (Reviewer #3).

The reviewers have discussed the reviews with one another and the Reviewing Editor has drafted this decision to help you prepare a revised submission.

Summary:

In this manuscript, Seol and colleagues use a combination of single-molecule and bulk biochemical assays together with simulations to determine DNA unwinding parameters for *E. coli* RecQ helicase. The authors propose that the core of RecQ helicase pauses during duplex unwinding, that the pauses depend on the bp stability and that the interaction of HRDC with released ssDNA amplifies the pauses. The authors also propose the mechanochemical/kinetic mechanism of the RecQ.

The reviewers agreed that the work is important and generally interesting, since it investigates the sequence-dependent behavior of RecQ helicase that is involved in controlling homologous recombination. Overall, the experiments are well designed, expertly performed and well controlled. The model describing pause amplification my HRDC is plausible, interesting and have important implications for the RecQ function in heteroduplex rejection and perhaps in other processes of DNA metabolism. The manuscript does a nice job of tying together new and published observations into a cohesive and interesting unwinding mechanism for RecQ. Several points, however, need to be addressed:

Essential revisions:

1) The authors state that they are monitoring homology sensing mechanism. However, the sequence used in the study did not contain any mismatch or non-homologous regions to contrast the case of homology. The authors can either do additional experiments to include such result (preferable) or they can tone down the claim by changing the title and corresponding text.

2) The reviewers agreed that RecQ-WT pauses as discussed in the article, but RecQ-WT also makes significant back steps (Figure 1B), because the position in bp does not only increase with the time, it also decreases. Authors should comment on this, whether it is dissociation of RecQ-WT or something else, such as backstepping, sliding or shuttling? Simulations of the duplex unwinding include only forward stepping and pauses, but no backstepping. Please elaborate.

Minor points:

1) It would be helpful to note segments in the traces for RecQ and RecQ-dH that are considered "pauses".

2) The authors note that "The roughly 20-fold difference in the time the enzyme requires to unwind the DNA at the longest pause duration sites in comparison to the average time required to unwind a single base pair, suggests that more than a single base pair is being opened by the enzyme during each kinetic step in the unwinding reaction." This requires further clarification.

3) What evidence is there to show that Na^+^ titration in Figure 3D is affecting base pair stability only. This experiment could be complicated by multiple ionic effects, including protein/DNA interactions.

4) in vivo RecQ does not act alone but instead works with other factors (SSB or RecJ) that could occlude the HRDC domain from the 5' unwound strand. The authors should discuss how competition between the HRDC domain and these other factors may play out in cells.

5) A recent crystal structure of A bacterial RecQ identified a guanine binding pocket on the surface of the helicase that is important for G-quadruplex unwinding (Voter et al., 2019). Could this pocket be important for pausing at GC-rich sites? Does it matter whether guanines are on the translocation strand versus the displaced strand?

6) When discussing mechanochemical coupling, please explain what do you mean when you say "ATP-gS incorporation"?

7) The evidence of sequence-dependent activity is suggested in Figure 2 through the histogram of dwell times and base-pair stability. It is confusing how dwell times peaks are paired with base-pair stability peaks in Figure 2A to generate Figure 2B. I suggest authors use a simpler method such as the correlation function between measured and calculated stability curve. The method of multiple gaussian fits combined with linear fits unnecessarily over processes the data. Figure 2—figure supplement 2A and B also shows a shift between indexed peaks. For example, peak #4 in Figure 2—figure supplement 2A is more than 30 bp away from peak #4 in Figure 2—figure supplement 2B. This is potentially a mistake, and this analysis method should be avoided. The histogram in Figure 2 should be also qualitatively described in terms what do the peaks mean.

8) The authors should show the variability between different observations of RecQ activity. For example, RecQ-dH in Figure 1B shows a trace that is 4 s long, however, that time can be anywhere between 2.5 and 5 s. Perhaps, the authors can superimpose several different traces that all start at 0 time, to show the actual variability between different instances.

---

## [Author Response]

Essential revisions:1) The authors state that they are monitoring homology sensing mechanism. However, the sequence used in the study did not contain any mismatch or non-homologous regions to contrast the case of homology. The authors can either do additional experiments to include such result (preferable) or they can tone down the claim by changing the title and corresponding text.

We thank the referees for suggesting this excellent idea to test the model with non-homologous DNA. The revised manuscript includes new data (Figure 6B) that demonstrates that a single mismatch located in a high GC region of the hairpin results in a significant decrease in pausing as compared with the intact DNA hairpin. This result confirms our model in which the inherent sequence-dependent pausing of RecQ core is amplified by the HRDC-ssDNA interaction and also suggests this non-linear amplification of pausing may be employed for suppressing illegitimate recommendation in vivo.

2) The reviewers agreed that RecQ-WT pauses as discussed in the article, but RecQ-WT also makes significant back steps (Figure 1B), because the position in bp does not only increase with the time, it also decreases. Authors should comment on this, whether it is dissociation of RecQ-WT or something else, such as backstepping, sliding or shuttling? Simulations of the duplex unwinding include only forward stepping and pauses, but no backstepping. Please elaborate.

We thank the referees for pointing out this subtle but important point of possible confusion. We agree with the referee’s assessment that RecQ-WT indeed not only pauses but also exhibits “shuttling” behavior, repetitive unwinding and rapid rewinding at high GC cluster region. We attribute this shuttling behavior to unwinding and annealing/backstepping of RecQ core while it is tethered to the DNA via its HRDC as proposed in Figure 6C. We also address the shuttling behavior in more detail in Figure 2—figure supplement 1. We found that two modes of reannealing occur during shuttling: sliding (fast reduction of DNA extension) or backtracking/translocating (occurring at a similar rewinding rate as the forward unwinding rate). Concerning RecQ dissociation, we consider it unlikely that RecQ-WT entirely dissociates from the DNA hairpin since this would result in complete closing of the DNA hairpin (explained in Figure 1B), which defines the end of the shuttling activity. Furthermore, the fact backsliding events end at characteristic locations, even at very low RecQ concentrations (as low 10 pM) suggests that it is unlikely that another RecQ molecule would be always bound at the stop location, lending further support to the model in which the rapid rezipping events correspond to backsliding of RecQ rather than disassociation of RecQ from the substrate.

The reviewers are correct that we do not simulate the WT RecQ trajectories. We instead focused on simulations of HRDC-deletion mutant that are much better described by relatively simple kinetic stepping schemes as detailed in the main text. As the reviewer points out, the behavior of the WT enzyme is significantly more complex than that of the HRDC deletion construct and, due to the highly stochastic, branched, and non-linear nature of these additional behaviors, they have proved difficult to characterize in sufficient detail to warrant including these effects in a full simulation of WT-RecQ. We can learn a great deal about the mechanism of unwinding and coupling of ATP to this process through the analysis and simulation of the HRDC-deletion construct data that does not exhibit complex back-tracking and shuttling behaviors.

Minor points:1) It would be helpful to note segments in the traces for RecQ and RecQ-dH that are considered "pauses".

We thank the reviewers for pointing out this oversight. In the revised manuscript the pausing locations are clearly indicated in Figure 1B.

2) The authors note that "The roughly 20-fold difference in the time the enzyme requires to unwind the DNA at the longest pause duration sites in comparison to the average time required to unwind a single base pair, suggests that more than a single base pair is being opened by the enzyme during each kinetic step in the unwinding reaction." This requires further clarification.

We thank the reviewers for pointing out this confusing statement. We agree that the statement is not clear and have rewritten the section to highlight the salient point more clearly. The point that we are trying to make is that, if RecQ-dH unwinds one base pair per each kinetic step, the largest difference between a single base-pair opening energy (G/C vs. A/T) is ~2.0 k_B_T, which corresponds to a ~7-fold increase in pausing duration. However, we observed a ~20-fold increase in the longest pause duration compared to the average unwinding rate, suggesting that more than a single base pair is being opened by the enzyme during each kinetic step. We have clarified this general point in the revised manuscript.

3) What evidence is there to show that Na^+^ titration in Figure 3D is affecting base pair stability only. This experiment could be complicated by multiple ionic effects, including protein/DNA interactions.

We agree with the reviewers’ concern with the possible pleotropic effects of Na^+^ concentration on RecQ-DNA interactions beyond altering the duplex stability and apologize that we did not make this point clear. Indeed, we found that the waiting times between events increased whereas the processivity (characterized by the mean number of unwound base-pairs before dissociation from DNA) decreased as a function of the monovalent salt concentration, indicating that the higher salt concentration affects the overall binding affinity of RecQ. The reduced unwinding rates in terms of increasing Na^+^ concentration, albeit being lower than expected, indicate that duplex unwinding is a rate-limiting step. However, possible pleotropic effects of Na^+^ concentration might not be well explained within the framework of our simple model. Therefore, we performed two additional experiments to more rigorously distinguish between the two unwinding models: Ensemble measurements in which the unwinding rate and kinetic step-size were obtained for substrates with different GC content (Figure 4), and direct step-size measurement using ATPγS (Figure 5). In the revised manuscript we have added a brief sentence to make clear that the Na^+^ dependent unwinding is consistent with, but not conclusive for, the multiple base-pair unwinding model due to possible pleotropic effects, but is confirmed by the results of the two subsequent additional measurements.

4) in vivo RecQ does not act alone but instead works with other factors (SSB or RecJ) that could occlude the HRDC domain from the 5' unwound strand. The authors should discuss how competition between the HRDC domain and these other factors may play out in cells.

We thank the reviewers for pointing out the possibility that SSB/RecJ interactions may compete with the HRDC for ssDNA. Whereas we do not know the details of how these interactions may affect or regulate RecQ activity in vivo, we speculate that the proximity of the HRDC to ssDNA effectively increases its local concentration thereby permitting the HRDC to out-compete other ssDNA binding proteins for binding newly melted ssDNA. Single-stranded DNA binding protein (SSB) may promote RecQ-mediated D-loop disruption activity as SSB can bind exposed ssDNA regions distal to RecQ to potentially prevent ssDNA from reannealing, thus slowing recombination processes relying on ssDNA annealing. Furthermore, as we recently demonstrated, the physical interaction between RecQ and SSB promotes disruption of SSBssDNA resulting in passive removal of SSB from ssDNA (NAR Vol. 45; Pages 11878–11890). RecJ is a 5’-3’ exonuclease that is known to be involved in the end resection step of recombination.

However it is unclear if RecJ is involved in the subsequent D-loop formation process.

Furthermore, RecJ prefers a 5’-ssDNA overhang and requires RecQ to access blunt-ended DNA as shown in the study by Morimatsu et al. (PNAS Vol 111 p E5133). For these reasons it remains unclear if, and to what extent, RecJ can bind the internal ssDNA region in competition with HRDC binding. To summarize, the reviewer raises very good points, but our data do not directly address the potential effects of SSB and/or RecJ binding to ssDNA in modulating or interfering with HRDC binding in vivo. We have added a short section in the Discussion section making the broader point that the model we are proposing relies on the HRDC binding to the displaced strand, which could be diminished or modulated by competition for ssDNA by other cellular factors including SSB and RecJ, but that the definitive test of these possibilities will require further experimental verification.

5) A recent crystal structure of A bacterial RecQ identified a guanine binding pocket on the surface of the helicase that is important for G-quadruplex unwinding (Voter et al., 2019). Could this pocket be important for pausing at GC-rich sites? Does it matter whether guanines are on the translocation strand versus the displaced strand?

We thank the reviewers for pointing out the possibility that SSB/RecJ interactions may compete with the HRDC for ssDNA. Whereas we do not know the details of how these interactions may affect or regulate RecQ activity in vivo, we speculate that the proximity of the HRDC to ssDNA effectively increases its local concentration thereby permitting the HRDC to out-compete other ssDNA binding proteins for binding newly melted ssDNA. Single-stranded DNA binding protein (SSB) may promote RecQ-mediated D-loop disruption activity as SSB can bind exposed ssDNA regions distal to RecQ to potentially prevent ssDNA from reannealing, thus slowing recombination processes relying on ssDNA annealing. Furthermore, as we recently demonstrated, the physical interaction between RecQ and SSB promotes disruption of SSBssDNA resulting in passive removal of SSB from ssDNA (NAR Vol. 45; Pages 11878–11890). RecJ is a 5’-3’ exonuclease that is known to be involved in the end resection step of recombination. However it is unclear if RecJ is involved in the subsequent D-loop formation process.

Furthermore, RecJ prefers a 5’-ssDNA overhang and requires RecQ to access blunt-ended DNA as shown in the study by Morimatsu et al. (PNAS Vol 111 p E5133). For these reasons it remains unclear if, and to what extent, RecJ can bind the internal ssDNA region in competition with HRDC binding. To summarize, the reviewer raises very good points, but our data do not directly address the potential effects of SSB and/or RecJ binding to ssDNA in modulating or interfering with HRDC binding in vivo. We have added a short section in the Discussion section making the broader point that the model we are proposing relies on the HRDC binding to the displaced strand, which could be diminished or modulated by competition for ssDNA by other cellular factors including SSB and RecJ, but that the definitive test of these possibilities will require further experimental verification.

6) When discussing mechanochemical coupling, please explain what do you mean when you say "ATP-gS incorporation"?

We apologize that the term”incorporation” was confusing and have replaced it with “binding”.

7) The evidence of sequence-dependent activity is suggested in Figure 2 through the histogram of dwell times and base-pair stability. It is confusing how dwell times peaks are paired with base-pair stability peaks in Figure 2A to generate Figure 2B. I suggest authors use a simpler method such as the correlation function between measured and calculated stability curve. The method of multiple gaussian fits combined with linear fits unnecessarily over processes the data. Figure 2—figure supplement 2A and B also shows a shift between indexed peaks. For example, peak #4 in Figure 2—figure supplement 2A is more than 30 bp away from peak #4 in Figure 2—figure supplement 2B. This is potentially a mistake, and this analysis method should be avoided. The histogram in Figure 2 should be also qualitatively described in terms what do the peaks mean.

We apologize for the lack of clarity and unnecessary confusion in describing how the peak positions in Figure 2B were correlated and we appreciate the reviewer’s comments concerning alternative approaches to establish the correlation between the pause locations and the duplex stability. We paired the positions based on the locations of the peaks determined in the base-pair energy and the dwell time histograms of RecQWT and RecQ-dH. The numbering indicated in Figure 2—figure supplement 2 simply indicated the indexing of the peaks obtained by the multipeak fitting routine rather than the correlation between peaks. To avoid possible confusion, we have changed the peak numbering in Figure 2—figure supplement 2 to correspond to the pairing in Figure 2B in the revised manuscript.

The dwell-time histograms represent the accumulated time that the duplex spends in each unwound state (bp unwound) in an unbiased manner. Thus, peaks in the dwell time histograms correspond to locations where RecQ pauses with high frequency and/or long durations. In principle the dwell time distribution and base-pair stability curves could be directly correlated, however the resulting correlation is less informative than the correlation we demonstrate among multiple peaks in the distributions. The correlation function displays a single peak at zero delay (see Author response image 1), consistent with the model, but this approach compresses all of the data to a single value. Demonstrating the near perfect correlation between individual peaks in the two distributions provides more compelling evidence of the strong correlation between base-pair energy and pausing than the single cross correlation peak at zero delay.

8) The authors should show the variability between different observations of RecQ activity. For example, RecQ-dH in Figure 1B shows a trace that is 4 s long, however, that time can be anywhere between 2.5 and 5 s. Perhaps, the authors can superimpose several different traces that all start at 0 time, to show the actual variability between different instances.

We thank the reviewers for this suggestion. We have included more unwinding traces of RecQ-dH in Figure 1B.